# The impact of introduction of the 10-valent pneumococcal conjugate vaccine on pneumococcal carriage in Nigeria

Aishatu L. Adamu [1,2,3] ✉, J. Ojal [1,2], Isa A. Abubakar[3], Kofo A. Odeyemi[4], Musa M. Bello[3], Christy A. N. Okoromah[5], Boniface Karia[1], Angela Karani[1], Donald. Akech[1], Victor Inem[4], J. Anthony G. Scott[1,2,7] & Ifedayo M. O. Adetifa[1,2,5,6,7]

Pneumococcal conjugate vaccines (PCVs) protect against invasive pneumococcal disease (IPD) among vaccinees. However, at population level, this protection is driven by indirect effects. PCVs prevent nasopharyngeal acquisition of vaccine-serotype (VT) pneumococci, reducing onward transmission. Each disease episode is preceded by infection from a carrier, so vaccine impacts on carriage provide a minimum estimate of disease reduction in settings lacking expensive IPD surveillance. We documented carriage prevalence and vaccine coverage in two settings in Nigeria annually (2016–2020) following PCV10 introduction in 2016. Among 4,684 rural participants, VT carriage prevalence fell from 21 to 12% as childhood (<5 years) vaccine coverage rose from 7 to 84%. Among 2,135 urban participants, VT carriage prevalence fell from 16 to 9% as uptake rose from 15 to 94%. Within these ranges, carriage prevalence declined with uptake. Increasing PCV10 coverage reduced pneumococcal infection at all ages, implying at least a comparable reduction in IPD.

In 2015, pneumococcal disease was estimated to cause ~300,000 deaths globally among children aged 1–59 months. Over 50% of these deaths occurred in Africa, and Nigeria alone accounted for nearly 50,000 of these pneumococcal deaths[1]. Between 2014 and 2016, in three geographically distinct phases, Nigeria introduced the 10-valent Pneumococcal Conjugate Vaccine (PCV10) in a three-dose schedule for infants aged 6, 10 and 14 weeks, without a catch-up campaign. Although PCV is the most expensive vaccine programme in the Nigerian portfolio, the country could not evaluate the impact of the vaccine programme on invasive disease or pneumonia due to lack of surveillance data.

Every episode of pneumococcal disease is preceded by infection from another infected person, normally a nasopharyngeal carrier[2]. Young children are the main reservoirs for carriage and have the highest number of effective contacts[3,4]. Consequently, a reduction in

carriage prevalence among young children is likely to reduce onward transmission and the incidence of disease proportionately across the population. Among vaccinated children, PCVs provide direct protection against both acquiring carriage and progressing to invasive disease following carriage of vaccine-serotypes (VTs)[2]. At the population level, PCVs provide indirect protection, regardless of vaccine status, by reducing everyone's exposure to new infections from VTs. This indirect effect is driven by the direct protection against carriage among vaccinees[5,6]. As vaccine coverage increases, VT carriage prevalence declines linearly due to direct protection among vaccinees and non-linearly due to indirect protection from the consequences of reduced VT transmission in the whole population[2,5].

In real-world settings, the indirect effects of PCVs account for most of the vaccine programme impact[2,7]. Consequently, some countries have tailored their PCV schedules to maximise indirect effects of a

[1]KEMRI-Wellcome Trust Research Programme, Kilifi, Kenya. [2]Department of Infectious Diseases Epidemiology, London School of Hygiene & Tropical Medicine, London, UK. [3]Department of Community Medicine, College of Health Sciences, Bayero University, Kano/Aminu Kano Teaching Hospital, Kano, Nigeria. [4]Department of Community Medicine and Primary Care, College of Medicine, University of Lagos, Lagos, Nigeria. [5]Department of Paediatrics and Child Health, College of Medicine, University of Lagos, Lagos, Nigeria. [6]Nigeria Centre for Disease Control, Abuja, Nigeria. [7]These authors jointly supervised this work: J. Anthony G. Scott, Ifedayo M. O. Adetifa. ✉e-mail: aladamu.cmed@buk.edu.ng

booster dose at the expense of marginal direct effects of additional primary doses in infancy. For example, in the UK, population protection is being achieved with only a single dose in infancy and a booster dose at 12 months[8]. A disadvantage of PCV introduction is replacement carriage by non-vaccine serotypes (NVTs) leading, to a varying extent, to serotype replacement disease[9,10]. However, in most settings, any increase in serotype replacement disease is small compared to the reduction in vaccine-type disease because non-vaccine types are generally less invasive[10].

In the absence of robust IPD surveillance and given the strong anticipation of indirect protection following PCV10 introduction, we set out to evaluate the impact of the Nigerian PCV programme using carriage prevalence as an endpoint[11]. In Nigeria, among children aged <5 years who were studied immediately after PCV10 introduction, from a rural and an urban setting, VT pneumococci accounted for 52 and 64% of all carriage, respectively[12]. We conducted annual carriage and vaccination coverage surveys in these same two sites, for 4 years following PCV10 introduction. We assessed changes in the prevalence of overall carriage (i.e. all pneumococci), and VT and NVT carriage separately and explored the relationship between changes in vaccination uptake and changes in VT carriage prevalence.

## Results

Including the baseline survey, reported above[12], we conducted five annual carriage surveys in the rural and four in the urban sites (Fig. 1) and recruited 4684 and 3653 participants, respectively. In the rural and urban sites, the proportion of eligible residents who consented to participate varied from 60–98% and 63–99%, respectively, across the sampling age groups and surveys (Supplementary Fig. 1 and Supplementary Table 1).

Participants in the rural site resided in larger households and more commonly reported living with ≥2 children aged <5 years, using solid fuel for cooking, and having a cough or runny nose in the preceding two weeks compared to their counterparts in the urban site (Table 1).

### Carriage prevalence

Table 2 shows the crude and age-standardised carriage prevalence stratified by survey. Among the age-standardised results, overall pneumococcal carriage prevalence was consistently high across all ages in all surveys at the rural site. At both sites, overall pneumococcal carriage prevalence and NVT carriage prevalence were higher in children aged <5 years compared to persons aged ≥5 years; VT carriage prevalence was also higher in children aged <5 years in the baseline surveys at both sites. The crude carriage prevalence (by sampled ages) is also illustrated in Supplementary Fig. 2.

### Changes in carriage prevalence

Overall carriage prevalence in the total population (all ages combined) remained unchanged across the surveys, in both settings (Tables 2 and 3). However, in the rural site (Table 2), overall carriage prevalence increased significantly among persons aged ≥5 years ($\chi^2$ test for trend, $p = 0.004$), and in the urban site (Table 3), overall carriage prevalence declined significantly among children <5 years ($\chi^2$ test for trend, $p < 0.0001$).

In the total population VT carriage prevalence steadily declined from 21 to 12% ($\chi^2$ test for trend, $p < 0.001$) in the rural site and from 16 to 9% ($\chi^2$ test for trend, $p < 0.001$) in the urban site. In the total population VT carriage prevalence steadily declined from 21 to 12% ($\chi^2$ test for trend, $p < 0.001$) in the rural site and from 16 to 9% ($\chi^2$ test for trend, $p < 0.001$) in the urban site. Among the total population sample, there was a significant trend for an increase in NVT carriage over the survey years in the rural site (Chi squared test for trend $p < 0.001$) and but not in the urban site (Chi squared test for trend $p = 0.36$).

For both age groups, VT carriage declined significantly across surveys in at each site ($\chi^2$ test for trend, $p < 0.001$ for all 4 trends). NVT carriage prevalence increased significantly in both age groups across surveys but only at the rural site ($\chi^2$ test for trend, $p < 0.001$).

Compared to the baseline survey, the adjusted age-standardised PR for VT carriage prevalence in the final survey was 0.52 and 0.53 (Table 4) among children <5 years and older persons, respectively, in Kumbotso (rural). The adjusted PRs were 0.31 and 0.60 among children <5 years and older persons, respectively, in Pakoto (urban). NVT carriage increased significantly in both age groups in Kumbotso, with adjusted PRs of 1.34 and 1.26 in children aged <5 years and persons ≥5 years, respectively. In Pakoto, serotype replacement carriage was significant only in those aged ≥5 years (adjusted PR 1.36, Table 4).

For children aged <5 years, the individual serotypes with the highest age-standardised prevalence in the final surveys were 6A (11.4%), 19F (5.5%) and 19A (5.4%), 11A (4.7%), 14, (4.4%) 16F (4.4%) and 23F (3.7%) in the rural site (Fig. 2 and Supplementary Table 2); and 19A (7.4%), 15B (4.6%), 6B (4.0%), 19F (3.9%), and 16F (3.7%) in the urban site (Fig. 2 and Supplementary Table 3). Among persons aged ≥5 years, in

## Timelines for Carriage and PCV10 coverage surveys

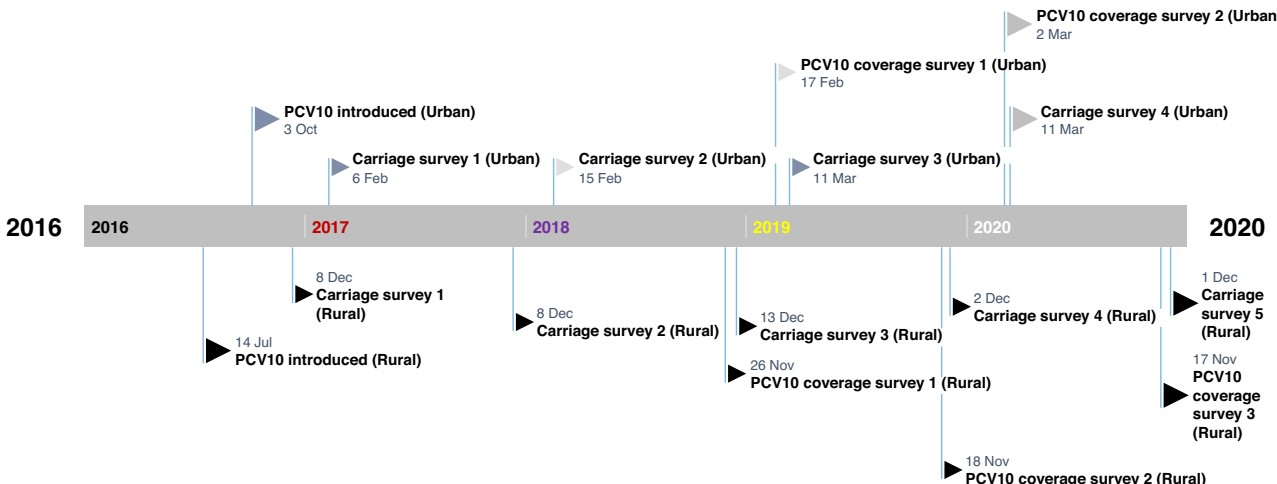

**Fig. 1 | Timelines for surveys in the two sites.** For each site, surveys were conducted around the same time. Note PCV10 coverage surveys started from 2018 onwards.

**Table 1 | Background characteristics of study participants of the carriage surveys**

| | N (%) Survey 1 (2016) | N (%) Survey 2 (2017) | N (%) Survey 3 (2018) | N (%) Survey 4 (2019) | N (%) Survey 5 (2020) |
|---|---|---|---|---|---|
| **Kumbotso (rural)** | | | | | |
| Total sample | 878 | 879 | 999 | 973 | 954 |
| Clinical history[a] | | | | | |
| Runny nose (%) | 714 (81) | 681 (77) | 900 (90) | 843 (87) | 727 (76) |
| Cough (%) | 450 (51) | 551 (63) | 687 (69) | 558 (57) | 487 (51) |
| Antibiotic use (%) | 65 (7) | 431 (49) | 510 (51) | 233 (24) | 202 (21) |
| Household composition | | | | | |
| Living with ≥2 aged <5 years (%) | 645 (73) | 469 (53) | 555 (56) | 619 (64) | 748 (78) |
| Sharing bed with ≥2 persons (%) | 729 (83) | 704 (80) | 882 (88) | 795 (82) | 857 (90) |
| Household cooking fuel | | | | | |
| Solid fuel (%) | 833 (95%) | 795 (90) | 959 (96) | 892 (92) | 850 (89) |
| Gas (%) | 12 (1%) | 19 (2) | 18 (2) | 38 (4) | 51 (5) |
| Kerosene (%) | 16 (2%) | 18 (2) | 5 (0.5) | 66 (0.6) | 3 (0.3) |
| Others (%) | 17 (2%) | 47 (5) | 17 (2) | 40 (4) | 46 (5) |
| Household size[b] | | | | | |
| All persons, median (IQR) | 9 (7–13) | 6 (3–10) | 6 (4–9) | 8 (6–10) | 9 (7–12) |

| | Survey 1 (2017) | Survey 2 (2018) | Survey 3 (2019) | Survey 4 (2020) | |
|---|---|---|---|---|---|
| **Pakoto (urban)** | | | | | |
| Total sample | 924 | 943 | 932 | 854 | N/A |
| Clinical history[a] | | | | | |
| Runny nose (%) | 238 (26) | 163 (17) | 106 (11) | 51 (6) | N/A |
| Cough (%) | 216 (23) | 122 (13) | 75 (8) | 32 (4) | N/A |
| Antibiotic use (%) | 145 (16) | 76 (8) | 39 (4) | 10 (1%) | N/A |
| Household composition | | | | | |
| Living with ≥2 aged <5 years (%) | 95 (10) | 81 (9) | 69 (7) | 53 (6) | N/A |
| Sharing bed with ≥2 persons (%) | 185 (20) | 212 (23) | 121 (13) | 121 (14) | N/A |
| Household cooking fuel | | | | | |
| Solid fuel (%) | 58 (6) | 38 (4) | 35 (4) | 11 (1) | N/A |
| Gas (%) | 326 (35) | 584 (62) | 713 (76) | 775 (91) | N/A |
| Kerosene (%) | 515 (56) | 238 (25) | 155 (17) | 29 (3) | N/A |
| Others (%) | 25 (3) | 83 (8) | 29 (3) | 39(5) | N/A |
| Household size[b] | | | | | |
| All persons, median (IQR) | 4 (3–5) | 5 (4–6) | 5 (4–6) | 5 (4–6) | N/A |

[a]History of any of the symptoms in the 2 weeks preceding the interview date.
[b]Including the participant.

the rural site (Supplementary Table 4), the most prevalent serotypes in the final surveys were 3, 34, 11A, 16F and 10A; in the urban site, the most prevalent serotypes were 11A, 3, 19A, 4, 23B and 38 (Fig. 2 and Supplementary Table 5).

In the rural site(Supplementary Tables 2 and 4), significantly increased prevalence odds (final vs baseline survey) were observed for serotypes 16F (OR 12.6) and 10A (11.6), among children aged <5 years, and for serotypes 19A (4.4), 16F (2.9), 10A (2.4), and 37(5.0) for persons aged ≥5 years. In the urban site (Supplementary Tables 3 and 5), NVT

replacement was significant for serotypes 19A (OR 2.3), 15B (2.6) and 16F (5.0) in children aged <5 years; there was no significant increases in individual NVTs among persons aged ≥5 years.

We compared the carriage prevalence of serotypes included in different PCV formulations (Supplementary Table 7) in the final survey among children <5 years old. The total carriage prevalences of all serotypes contained in the Serum Institute of India 10-valent PCV (SII-PCV), 13-valent PCV (PCV13), 15-valent PCV (PCV15) and 20-valent PCV (PCV20) were 54%, 61%, 62% and 68%, respectively, in the rural site and 50%, 53%, 53% and 60%, respectively, in the urban site.

### PCV10 vaccine coverage
We assessed the PCV10 vaccination status of 2165 children (aged <5 years) in the rural site and 1313 children in the urban site. We accepted either written evidence of vaccination or the caregiver's recall. The average proportion of children for whom the caregivers had retained their vaccination card was 70% in the rural site (52% in 2018; 77% in 2019; and 90% in 2020) and 80% in the urban site (70% in 2019 and 91% in 2020). Figure 3A shows the annual proportions of children aged <5 years who had received at least two doses of PCV10. PCV10 coverage (≥2 doses) increased steadily from 7% in 2016 to 84% in 2020, in the rural site; and from 15% in 2017 to 94% in 2020, in the urban site.

### Relationship between PCV10 coverage and VT carriage
Within the range of PCV10 coverage observed in children, the ecological relationship between PCV10 coverage and the prevalence of VT carriage (Fig. 3B) shows a linear decline for older persons aged ≥5 years in both settings (gradient −0.09 (95% CI −0.13 to −0.04) in Kumbotso; −0.07 (95% CI −0.10 to −0.04) in Pakoto). For children aged <5 years, a log-linear model had a better fit to the data (Supplementary Fig. 3) which show a steep decline in VT carriage prevalence associated with a small increase in PCV coverage towards 20% followed by slower gains as coverage increases further.

### Discussion
The aim of this study was to evaluate the introduction of a new, expensive vaccine programme in Nigeria using an inexpensive proxy measure of impact, vaccine-type nasopharyngeal carriage. Over five years, in a rural setting (Kumbotso) in northern Nigeria, the proportion of children aged <5 years who were vaccinated increased from 7 to 84%. During the same period, the age-standardised population prevalence of VT carriage fell from 21 to 12%, giving an adjusted prevalence ratio of 0.52 or a VT carriage reduction of 48%. Over three years, in an urban setting (Pakoto) in southern Nigeria, the proportion of children vaccinated increased from 15 to 94%. During the same period, the age-standardised population prevalence of VT carriage fell from 16 to 9%, giving an adjusted PR of 0.34 or a reduction in carriage of 66%. In both settings, we observed a decrease in VT carriage prevalence among children and older persons as vaccine coverage among children <5 years accumulated over time. For older persons (aged ≥5 years) this relationship was approximately linear representing a reduction in VT carriage prevalence of 1.4–1.5% for every 20% increase in vaccine coverage among children in the same setting.

Although carriage is only a proxy, we can use it to infer the impact of PCV10 on disease rates in these settings. A reduction in carriage prevalence will produce a proportionate reduction in the number of carriers each person contacts, reducing the incidence of carriage acquisition and the incidence of all pneumococcal diseases commensurately. A reduction in VT carriage prevalence of 66% at all ages in Pakoto is likely to translate into a reduction in the incidence of all VT pneumococcal disease of at least 66% at all ages. This estimate considers only the indirect effect of the programme, but it is, in itself, a very significant public health gain. Direct effects cannot be estimated from these surveys, but in an individually-randomised controlled trial of PCV9 in The Gambia, vaccine efficacy against VT IPD was 77%[13].

**Table 2 | Crude and age-standardised[a] prevalence (and 95% CI) of overall, non-vaccine serotype (NVT) and vaccine serotype (VT) pneumococcal carriage stratified by age group and survey in the rural site**

| Survey | N | Overall carriage | | VT carriage | | NVT carriage | |
|---|---|---|---|---|---|---|---|
| | | Crude | Age-standardised | Crude | Age-standardised | Crude | Age-standardised |
| Kumbotso (rural) | | | | | | | |
| All ages | | | | | | | |
| Survey 1 (2016) | 872 | 74 (71–77) | 68 (65–71) | 26 (22–28) | 21 (18–24) | 48 (45–52) | 47 (43–51) |
| Survey 2 (2017) | 879 | 74 (71–77) | 71 (67–74) | 18 (16–21) | 16 (14–19) | 55 (52–59) | 54 (51–58) |
| Survey 3 (2018) | 999 | 77 (74–80) | 77 (74–79) | 16 (14–19) | 16 (13–18) | 60 (57–64) | 61 (58–64) |
| Survey 4 (2019) | 976 | 77 (74–79) | 74 (71–77) | 15 (13–17) | 13 (11–15) | 61 (59–65) | 60 (57–64) |
| Survey 5 (2020) | 953 | 78 (75–80) | 74 (71–77) | 14 (12–17) | 12 (10–14) | 63 (61–67) | 61 (58–65) |
| <5 years | | | | | | | |
| Survey 1 (2016) | 296 | 92 (88–94) | 91 (88–94) | 42 (37–48) | 41 (35–46) | 50 (44–56) | 50 (45–56) |
| Survey 2 (2017) | 264 | 93 (89–95) | 92 (89–96) | 30 (25–36) | 30 (25–36) | 63 (57–68) | 62 (56–68) |
| Survey 3 (2018) | 304 | 93 (89–95) | 92 (90–95) | 25 (21–30) | 25 (20–30) | 68 (62–73) | 67 (62–72) |
| Survey 4 (2019) | 365 | 91 (88–94) | 91 (88–94) | 21 (17–26) | 22 (17–26) | 70 (65–75) | 69 (64–74) |
| Survey 5 (2020) | 333 | 89 (85–92) | 88 (84–91) | 22 (18–27) | 22 (18–27) | 67 (61–72) | 65 (60–71) |
| >5 years | | | | | | | |
| Survey 1 (2016) | 576 | 65 (60–68) | 62 (58–66) | 17 (14–20) | 16 (13–19) | 48 (43–52) | 46 (42–50) |
| Survey 2 (2017) | 615 | 66 (62–64) | 65 (61–69) | 13 (11–16) | 13 (10–16) | 53 (49–56) | 52 (48–56) |
| Survey 3 (2018) | 695 | 70 (67–74) | 73 (69–76) | 13 (10–15) | 13 (11–16) | 57 (54–61) | 59 (55–63) |
| Survey 4 (2019) | 611 | 68 (64–71) | 69 (66–73) | 11 (9–14) | 11 (9–14) | 57 (53–61) | 58 (54–62) |
| Survey 5 (2020) | 620 | 72 (69–76) | 70 (66–74) | 10 (8–13) | 9 (7–11) | 62 (58–66) | 61 (57–64) |

[a]Standardised using the respective population structures of the two study sites taken from population models of the Nigerian census[37].

**Table 3 | Crude and age-standardised[a] prevalence of overall, non-vaccine serotype (NVT) and vaccine serotype (VT) pneumococcal carriage stratified by age group and survey in the urban site**

| Survey | N | Overall carriage | | VT carriage | | NVT carriage | |
|---|---|---|---|---|---|---|---|
| | | Crude | Age-standardised | Crude | Age-standardised | Crude | Age-standardised |
| Pakoto (urban) | | | | | | | |
| All ages | | | | | | | |
| Survey 1 (2017) | 919 | 50 (47–53) | 40 (36–43) | 22 (19–24) | 16 (13–18) | 29 (25–31) | 24 (21–27) |
| Survey 2 (2018) | 941 | 52 (49–55) | 51 (47–54) | 15 (13–18) | 14 (12–17) | 37 (34–40) | 36 (33–39) |
| Survey 3 (2019) | 932 | 47 (44–50) | 44 (41–48) | 12 (10–14) | 11 (9–14) | 35 (32–38) | 33 (30–36) |
| Survey 4 (2020) | 851 | 40 (36–43) | 39 (36–42) | 9 (7–11) | 9 (6–10) | 31 (28–34) | 31 (28–34) |
| <5 years | | | | | | | |
| Survey 1 (2017) | 335 | 78 (73–82) | 77 (72–81) | 38 (33–43) | 36 (31–42) | 40 (35–45) | 40 (35–45) |
| Survey 2 (2018) | 244 | 70 (64–76) | 70 (65–76) | 23 (18–29) | 23 (18–29) | 47 (41–53) | 47 (41–54) |
| Survey 3 (2019) | 243 | 70 (64–75) | 69 (63–75) | 19 (15–25) | 19 (14–24) | 51 (44–57) | 50 (43–56) |
| Survey 4 (2020) | 185 | 52 (45–59) | 53 (46–61) | 12 (8–17) | 12 (7–17) | 40 (33–47) | 41 (34–49) |
| ≥5 years | | | | | | | |
| Survey 1 (2017) | 584 | 34 (31–38) | 32 (28–36) | 13 (10–15) | 12 (9–15) | 22 (19–25) | 20 (17–24) |
| Survey 2 (2018) | 697 | 46 (42–50) | 47 (43–50) | 12 (10–15) | 13 (10–15) | 34 (30–37) | 34 (30–38) |
| Survey 3 (2019) | 689 | 39 (36–43) | 40 (36–43) | 9 (7–12) | 10 (8–12) | 30 (26–33) | 29 (26–33) |
| Survey 4 (2020) | 666 | 36 (33–40) | 36 (33–39) | 8 (6–10) | 7 (5–9) | 29 (25–32) | 29 (25–32) |

[a]Standardised using the respective population structures of the two study sites taken from population models of the Nigerian census[37].

Therefore, even among the 34% of new pneumococcal infections that have not been potentially averted by indirect effects in Pakoto, the risk of developing disease will still be attenuated (by 77%) if the infected child has been vaccinated with PCV10, as most have.

This concept of additional gains from indirect vaccine effects is substantiated by the results from other settings. In Kilifi, Kenya, for example, a 74% decline in VT carriage prevalence among children aged <5 years was associated with a 92% decline in VT IPD in this age group[14]. In Sao Paulo, Brazil, a 91% decline in VT carriage prevalence among toddlers aged 12–23 months was associated with an 83–87% decline in VT IPD in children across the whole age range <5 years[15,16].

The decline in VT carriage prevalence in Nigeria was accompanied by an increase in NVT carriage prevalence among children in Kumbotso (rural) and among older persons in both settings, with adjusted prevalence ratios of 1.26–1.34. In Kenya, the 74% decline in VT carriage prevalence was accompanied by a 1.71-fold increase in NVT carriage prevalence, though there was no significant rise in serotype replacement disease[14]. Non-vaccine serotypes with high frequency in the final surveys in children <5 years were 6A, 19A, 11A, 15B, and 16F. The first two are contained in the alternative PCV10 manufactured by Serum Institute of India, and 11A and 15B are contained in the PCV20 recently licensed for adult use[17,18]. This NVT distribution suggests that if

**Table 4 | Prevalence ratios (PR), and 95% CI, showing changes in overall, non-vaccine serotype (NVT), and vaccine serotype (VT) carriage stratified by age and site**

| | Overall carriage | | VT carriage | | NVT carriage | |
|---|---|---|---|---|---|---|
| | Crude PR | Adjusted age-standardised PR[a] | Crude PR | Adjusted age-standardised PR[a] | Crude PR | Adjusted age-standardised PR[a] |
| PR for carriage in the final survey compared to the baseline survey[b] | | | | | | |
| Kumbotso (rural)[c] | | | | | | |
| All ages | 1.06 (1.00–1.11) | 1.00 (0.95–1.05) | 0.55 (0.45–0.67) | 0.52 (0.43–0.64) | 1.32 (1.22–1.44) | 1.30 (1.19–1.42) |
| <5 years | 0.97 (0.82–1.14) | 0.97 (0.92–1.02) | 0.52 (0.41–0.67) | 0.52 (0.41–0.67) | 1.34 (1.17–1.54) | 1.34 (1.17–1.54) |
| ≥5 years | 1.12 (0.97–1.28) | 1.06 (0.97–1.14) | 0.58 (0.43–0.78) | 0.53 (0.39–0.72) | 1.31 (1.18–1.46) | 1.26 (1.12–1.40) |
| Pakoto (urban)[d] | | | | | | |
| All ages | 0.79 (0.71–0.88) | 0.72 (0.65–0.80) | 0.40 (0.31–0.51) | 0.34 (0.26–0.45) | 1.09 (0.95–1.26) | 1.03 (0.89–1.20) |
| <5 years | 0.67 (0.58–0.78) | 0.68 (0.58–0.79) | 0.32 (0.21–0.48) | 0.31 (0.20–0.48) | 1.01 (0.81–1.25) | 1.02 (0.82–1.28) |
| ≥5 years | 1.05 (0.91–1.22) | 1.07 (0.90–1.28) | 0.61 (0.44–0.86) | 0.60 (0.41–0.87) | 1.30 (1.07–1.58) | 1.36 (1.10–1.69) |

[a]Adjusted for symptoms of upper respiratory tract infection in past 2 weeks, living with ≥2 children aged <5 years, and age-standardised to the respective age distribution of study sites.
[b]PR = prevalence ratios comparing each survey compared to the baseline (first) survey.
[c]Five surveys (2016–2020).
[d]Four surveys (2017–2020).

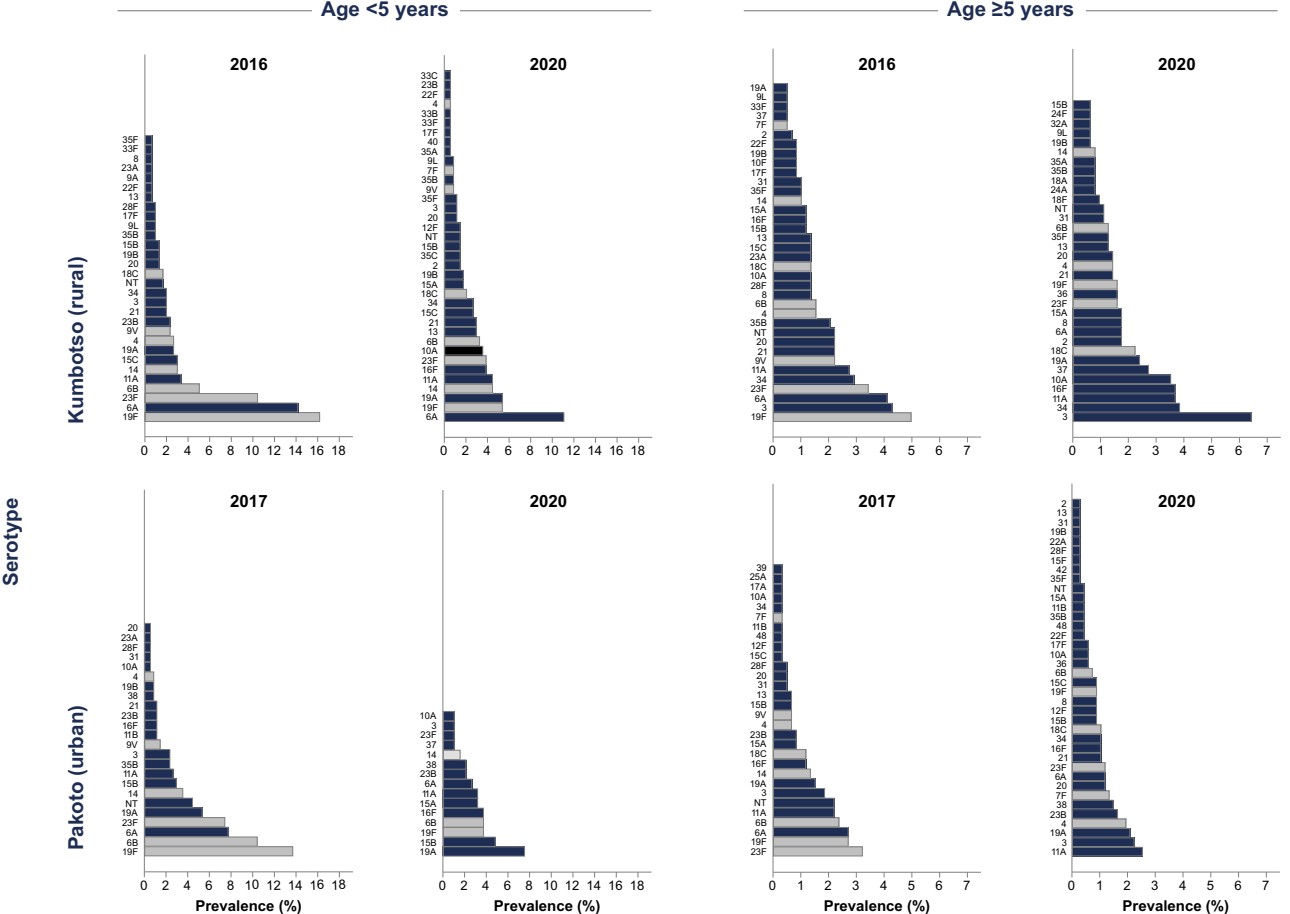

**Fig. 2 | Serotype-specific carriage prevalence per survey stratified by age group.** Distribution and ranking of serotypes in carriage (serotypes with >1 isolate) among children aged <5 years and persons ≥5 years by vaccine-type (greyscale bars – vaccine-serotypes, navy blue bars – non-vaccine serotypes) in the baseline and final surveys. Note the differences in scale in graphs by age.

serotype replacement disease becomes problematic, it may be controlled by wider valency vaccines. However, the relevance of serotype replacement carriage is dependent on the inherent invasiveness of the serotypes increasing in prevalence[19–21] which can only be ascertained from linked studies of carriage and IPD[14,20,22].

The study findings need to be interpreted in light of several practical constraints. The study began more than four months after

PCV10 introduction, and at the baseline survey, an estimated 7–15% of children aged <5 years had already been vaccinated. Had the baseline survey pre-dated PCV10 introduction, the measured impact may have been larger. The evaluation is a 'before-after' study which is susceptible to confounding by secular trends in VT carriage prevalence. It is difficult to control for this possibility in retrospect. Nonetheless, it is unlikely that secular trends alone could account for so large an effect

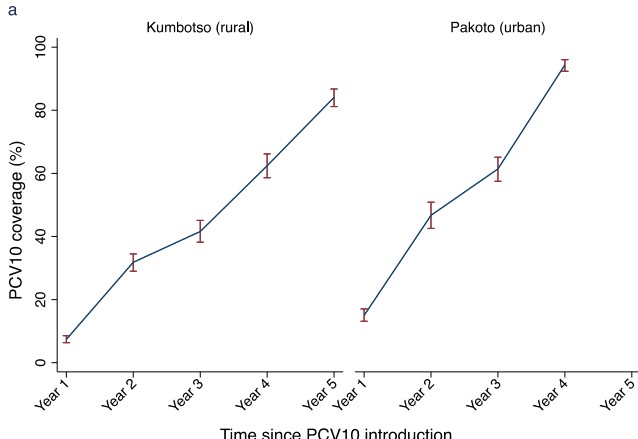

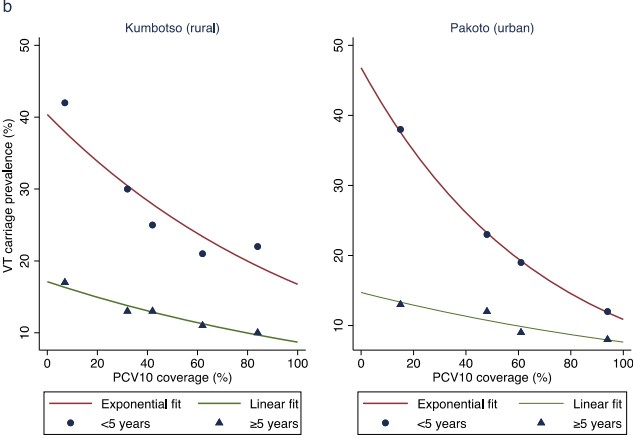

**Fig. 3 | Coverage of PCV10 and its relationship to VT carriage. a** (top) Annual Coverage of two doses of PCV10 among children aged < 5 years. Year 1 represents the year of PCV10 introduction. PCV10 coverage values for Year 3 to Year 5 were assessed directly among 817, 655 and 693 children in Kumbotso, and for Year 3 and 4 among 652 and 661 children in Pakoto. PCV10 coverage values for Year 1 and Year 2 were estimated using a birth-cohort analysis of children observed during Years 3–5 (among 2165 and 1140 children in Kumbotso, and 1,313 and 568 children in Pakoto). Error bar = 95% confidence interval CI). **b** (bottom). Relationship between Vaccine serotype (VT) carriage prevalence and PCV10 coverage. Scatter graph of log-linear regression among children aged <5 years and linear regression among persons ≥5 years of VT carriage prevalence against PCV10 coverage for each of the 9 surveys, stratified by age of carrier and shown separately for the Kumbotso (rural) site and Pakoto (urban) site. The lines for children (aged <5 years) are exponential fits (log-linear regression) and the lines for the older persons (age ≥5 years) are arithmetic (linear regression). Values from the log-linear regression among children are exponentiated and shown here on the non-log (arithmetic) scale.

size on VT carriage. The study design did, however, control for seasonal variation in pneumococcal carriage[23], as the surveys were done at the same time each year.

Vaccination coverage surveys were only introduced in 2018, and we inferred the coverage estimates for young children prior to 2018 from the coverage results among older children. Despite random selection and a study of adequate size, the coverage data contain internal inconsistencies; for example, in Pakoto, the rise in coverage in Year 3 (2019) was >40% and yet only ~20% of children aged <5 years were eligible to be vaccinated in that year. This may implicate poor recall of vaccination among caregivers of older children sampled in 2019. Vaccination coverage is notoriously difficult to ascertain[24]. Therefore, the ecological relationship we observe between coverage and VT carriage in older persons should be interpreted with some caution.

For practical reasons we selected two markedly different sites to represent the broad environmental and socio-demographic differences in Nigeria. However, we do not consider these sites to be wholly representative of all settings in Nigeria. Households in the rural site (Kumbotso) from northern Nigeria were larger, had more children and generally used solid cooking fuel. Households in the urban site (Pakoto) from southern Nigeria were smaller, had substantially fewer children and generally used gas and kerosene for cooking. At baseline, VT carriage prevalence was higher in the rural setting at all ages but, paradoxically, vaccine impact was greater in the urban setting, at least among children <5 years old; adjusted prevalence ratios were 0.52 in Kumbotso and 0.31 in Pakoto. This differential impact may be attributable to the steeper rise in PCV10 coverage among children aged <5 years in Pakoto. Alternatively, the lower density of children in urban households may imply a lower force of infection. A high force of infection has been proposed as an important cause of residual VT carriage in mature vaccine programmes in Africa[25], and in Kumbotso, VT carriage prevalence reaches its nadir at 22% in years 2019/2020, compared to 9% in Pakoto in 2020.

Hence, the impact of the vaccine on carriage prevalence is likely to be affected by several additional factors; the baseline serotype distribution, age-specific carriage prevalence, demography, the contact patterns of the community, the probability of transmission at each contact and the duration of carriage and of vaccine-induced immunity[26–28]. The age structure of the vaccinated population is also influential; for example, a catch-up campaign for children aged <5 years in Kenya elicited a 64–66% reduction in VT carriage prevalence at all ages within six months of PCV10 introduction[29]. The full interaction of these effects can only be understood within a formal framework, such as a dynamic transmission model. Even here, the accuracy of predicting disease depends on a clear understanding of the risk of disease per episode of carriage for both VTs and NVTs[30,31]. The full spectrum of data required to parameterise such a model is not currently available for Nigeria.

Among children, we found that VT carriage declines exponentially with a large reduction in VT carriage prevalence observed at low levels of increasing PCV10 uptake. In an ecological analysis in Australia, 73% of VT-IPD cases were estimated to have been prevented by approximately 50% vaccine uptake of PCV7 [32], which lends credence to the hypothesis that indirect effects may begin at relatively low levels of uptake. It is also possible that our data are capturing the dynamic stage of a complex polynomial effect, and the exponential fit works only within the coverage range we explored. Although both direct and indirect effects are expected in children, changes are mostly driven by the latter, which supports the non-linear effect observed. Given that the impact on adult carriage is entirely attributable to indirect effects, we would expect the same function should be observed in older people. The arithmetic decline we observed in this population is, therefore, difficult to explain.

We restricted our study to detect a single serotype in each swab despite abundant evidence supporting multiple serotype colonisation in children[32]. The dynamics and clinical importance of multiple serotypes in nasopharyngeal carriage are not fully understood[19,33]. Nonetheless, sampling a single strain per child provides a valid estimate of the distribution of serotypes colonising the population of children in these areas.

The measurable impact on VT carriage reported here should reassure immunisation policymakers and service providers in Nigeria that, in settings with similar baseline epidemiology and comparable vaccine coverage across the country, PCV10 is bringing about population protection through its indirect effect. This protection is likely to have reduced the incidence of pneumococcal disease among all ages by 48–66%, depending on the setting. Among the majority of children aged <5 years who have now received a course of PCV10, this indirect effect will have been augmented by direct effects that are likely to be

very strong. The decline in VT carriage prevalence as PCV10 coverage increases among children <5 years suggests that, in settings with suboptimal coverage, efforts to improve coverage will yield significant reductions in carriage and transmission and, therefore, disease incidence.

## Methods

### Study design and participants

We conducted annual cross-sectional carriage surveys in Kumbotso, Kano State and Pakoto, Ogun State (Fig. 1). The sites were purposively selected to represent a rural and urban setting, respectively. We did four surveys (2017–2020) in the rural site and three (2018–2020) in the urban site. PCV10 was introduced in Kumbotso in July 2016 and in Pakoto in October 2016 with a schedule of three primary doses (3p + 0) at ages 6, 10 and 14 weeks and no booster. There was no formal catch-up campaign for children aged ≥12 months. From 2018 onwards, we conducted annual vaccine coverage surveys in both sites simultaneously with all carriage surveys. The target population for the carriage and vaccine coverage surveys was defined as residents living within 10 km of the Kumbotso and Pakoto Comprehensive Primary Health Care Centres, respectively. Baseline carriage surveys were conducted in December 2016 (rural) and February 2017 (urban), four to five months after PCV10 was introduced, and have already been published[12]. They are included in this analysis as the reference baseline.

Carriage surveys were seasonally restricted at each site; November/December for four years (2017–2020) in the rural site and February/March for three years (2018–2020) in the urban site (Fig. 1). Carriage surveys targeted all ages, and each annual sample was independent of all other samples. PCV10 coverage surveys targeted children aged <5 years who were age-eligible to have received PCV10 at the date of the baseline carriage survey. Each annual PCV10 coverage sample was selected independently of prior samples.

Having selected representative study areas, we used a two-stage sampling design. In the first stage, we selected households using simple random sampling. To obtain a sampling frame, we conducted a census of all households in the catchment area before each survey. We selected separate samples of households for the carriage and PCV10 coverage surveys. If the household was known to be occupied, but there was no one at home, we revisited it later. If the house was non-residential, unoccupied, or empty, we chose the next household on the list.

In the second stage of sampling for the carriage surveys, we randomly selected one participant per household drawn from a specific age-stratum. We recruited participants in ten age strata (<1, 1–2, 3–4, 5–9, 10–14, 15–19, 20–39, 40–49, 50–59, and ≥60 years), starting with the lowest and moving upwards, from household to household, until we had recruited one participant per age group and then we restarted the process. If there was no participant in a particular age group in the household or if the targeted individual declined to participate, we selected the next age group in sequence and then looked for the missed age group in the next household.

The baseline surveys sampled the same defined catchment areas at all ages using a convenience sample of volunteers, recruited at the two health centres, recruited by community outreach[12]. For the baseline carriage surveys (2016/2017)[12], the sample size was set at 1000 participants to achieve a desired precision; given a VT carriage prevalence of 22–26% in this survey, we estimated a prevalence reduction of 50% could be detected with a power of 0.90 if the follow-up surveys were also 1000 in size. Therefore, we targeted to recruit 100 participants in each of the ten age groups.

In the second stage of sampling for the PCV10 coverage survey, we recruited all eligible children per selected household. A sample size of at least 639 children per site per survey was sufficient to estimate coverage of the second dose of PCV of 50% with a 5% precision (i.e., a coverage of 45–55%), assuming at least two eligible children per

household, an intra-class coefficient (ICC) of 0.33 (as recommended by WHO[34]) and an 80% probability of response or participation[35]. Targeting a vaccination coverage of 50% allowed the estimation of the largest possible sample size required.

### Procedure

Sociodemographic and clinical information was obtained from carriage survey participants using an interviewer-administered questionnaire. Nasopharyngeal swabbing, transport, storage and culture were done according to WHO-recommended standards[36]. We collected one swab specimen per participant from the posterior wall of the nasopharynx using nylon-tipped flexible flocked swabs (FloQS-wabs®). Swabs were transported to the laboratory within 8 h of collection in skimmed milk-tryptone-glucose-glycerine (STGG) on ice packs in a cold box and were stored at −80 °C to −55 °C before shipping on dry ice to the KEMRI-Wellcome Trust Research Programme (KWTRP), Kilifi, Kenya. In Kilifi, swabs were stored at −80 °C until they were thawed and cultured on blood agar with 5 μg/ml gentamicin.

We identified pneumococci by α-haemolysis and optochin sensitivity testing. For optochin-resistant isolates (zone of inhibition <14 mm diameter), we used bile solubility testing to confirm *S. pneumoniae*. For serotyping, we selected one colony per plate from the dominant colony morphology. We identified serotypes using latex agglutination confirmed by Quellung Reaction. For isolates with inconclusive serotyping, we confirmed species and serotype by polymerase chain reaction (PCR) for autolysin (*lytA*) and capsular locus genes, respectively[36].

For the PCV10 coverage survey, we obtained the PCV10 vaccination status of each child in the household, including doses and dates received from the vaccination cards or caregiver recall, through household interviews of caregivers.

### Statistical analysis

**Carriage surveys.** We calculated the total (all ages) and age-stratified prevalence of overall carriage (all pneumococci), VT pneumococci, and NVT pneumococci for each survey year. Vaccine serotypes (VT) were those contained in the vaccine introduced locally (PCV10 – serotypes 1, 4, 5, 6B, 7F, 9V, 14, 18C, 19F and 23F). Any other serotype, including non-typeable isolates, was classified as NVT. We recalculated VT prevalence for four other commercially licensed PCVs (Supplementary Table 8). We standardised crude prevalence estimates to the population age structure of Kumbotso (for rural) and Ifo and Ado-Ota (for urban) Local Government Areas (LGAs). These were obtained from 2019 population models of the 2016 Nigerian census data[37].

We assessed changes in carriage prevalence across the survey years using Chi-square test for trend. To derive prevalence ratios (PRs) comparing the last survey with the first, we modelled carriage prevalence using log-binomial regression or Poisson regression with robust standard errors when the models failed to converge. We adjusted PRs for exposure variables independently associated with carriage and survey year at $p < 0.1$ which included: living with children aged <5 years and a history of cough and runny nose in the preceding two weeks. We also adjusted for the stratified sampling method by (probability) weighting age-specific PRs by the local population age structure, as above, obtained from the Nigerian census data[37]. We calculated PRs for the total population (all ages), for children aged <5 years and for persons aged ≥5 years.

**Vaccination coverage surveys.** The purpose of the coverage survey was to infer population immunity, not to evaluate programme effectiveness. Therefore, we estimated PCV10 coverage in each survey year (2018–2020) as the proportion of children aged <5 years (regardless of age-eligibility) who received two doses of PCV10 irrespective of timing and age of receipt. In addition, because we did not conduct PCV10 coverage surveys in the early period (2016–2017), we used a birth

cohort analysis to estimate the PCV10 coverage of children aged <5 years retrospectively from the data collected in 2018–2020.

**Relationship between PCV10 coverage and VT carriage.** Within the range of vaccine coverage observed, we analysed a simple ecological association between population-level PCV10 coverage in children aged <5 years and VT carriage, in both children aged <5 years and persons aged ≥5 years, using linear regression. We considered a non-linear relationship between PCV10 coverage and VT carriage using a log-linear model and compared the fit of linear to the log-linear model graphically. We also examined this non-linear relationship by comparing the models using the Akaike Information Criterion (AIC). A lower value of AIC is a better fit model. To allow direct comparison of AIC values from the linear and log-transformed model, we adjusted the AIC of the log-linear model by adding the following quantity[38]:

$$2 \times \text{sum}(\log(\text{VT carriage})) \qquad (1)$$

We did all the analysis separately for each site with Stata® version 15.1(College Station, TX, USA).

### Reporting summary

Further information on research design is available in the Nature Portfolio Reporting Summary linked to this article.

## Data availability

The authors declare that data supporting the findings of this study are available within the paper and its supplementary information files (Supplementary Data). Additional data requests can be made to the KEMRI-Wellcome Trust Research Programme Data Governance Committee (dgc@kemri-wellcome.org).

## Code availability

Data were analysed using Stata® version 15.1 (College Station, Texas, USA).

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

## Acknowledgements

This work was funded by Wellcome Trust [214320 - JAGS] and NIHR Global Health Research Unit on Mucosal Pathogens [2268427 - IMOA]. A.L.A. is funded by the DELTAS Africa Initiative [DEL-15-003]. The DELTAS Africa Initiative is an independent funding scheme of the African Academy of Sciences (AAS)'s Alliance for Accelerating Excellence in Science in Africa (AESA) and supported by the New Partnership for Africa's Development Planning and Coordinating Agency (NEPAD Agency) with funding from the Wellcome Trust [107769/Z/10/Z] and the UK government. The views expressed in this publication are those of the author(s) and not necessarily those of AAS, NEPAD Agency, Wellcome Trust or the UK government. I.M.O.A. is funded by the United Kingdom's Medical Research Council and Department For International Development through the African Research Leader Fellowship (MR/S005293/1) and by the NIHR-MPRU at UCL (grant 2268427 LSHTM). J.A.G.S. is funded by a Wellcome Trust Senior Research Fellowship (214320) and the NIHR Health Protection Research Unit in Immunisation. J.O. is funded by the NIHR Global Health Research Unit on Mucosal Pathogens (16/136/46).

## Author contributions

A.L.A., I.M.O.A., J.A.G.S. and J.O. contributed to study concept and design. A.L.A. led the fieldwork with input from D.A., A.K., M.M.B., I.A.A., C.A.N.O., V.I. and K.O.; A.K. led the laboratory work. B.A. oversaw the curation and management of data. A.L.A. performed all statistical analyses with input from J.O., I.M.O.A. and J.A.G.S.; A.L.A. wrote the first draft. A.L.A., B.K., I.M.O.A. and J.O. had direct access to and have verified the underlying data reported in the manuscript. All authors contributed to critical revision of the manuscript for intellectual content, and had final responsibility for the decision to submit for publication.

## Competing interests

The authors declare no competing interests.

### Ethical approval

Ethics approval for study was granted by the Research Ethics Committees of Aminu Kano Teaching Hospital (NHREC/21/08/2008/AKTH/EC/2165), Kano State Ministry of Health (MOH/OFF/797/T.I/596), Lagos University Teaching Hospital (ADM/DCST/HREC/APP/10300); the Kenya Medical Research Institute's Scientific and Ethical Review Unit (SERU 3350); and by the London School of Hygiene and Tropical Medicine Observational/Interventions Research Ethics Committee (Ref. 11670).

### Informed consent

Written informed consent was obtained from participants/guardians.
