## [Peer Review File · Nature Communications]

The impact of introduction of the 10-valent pneumococcal conjugate vaccine on pneumococcal carriage in NigeriaREVIEWER COMMENTS

Reviewer #1 (Remarks to the Author):

Reviewer

Nature Communications Manuscript: NCOMMS-22-1011

Title: The impact of introduction of the 10-valent pneumococcal conjugate vaccine on pneumococcal carriage in Nigeria

This is a manuscript by Adamu et al., reporting the findings from an assessment of population-level impact of PCV introduction in rural and urban sites in Nigeria on pneumococcal carriage and PCV10 vaccination coverage. This a well written paper presenting findings from a country with limited research on the impact of PCV on pneumococcal carriage and disease. Findings are relevant to inform development of optimal PCV implementation strategies in low- and middle-income country (LMIC) settings.

Though I am not a certified statistician, the analysis was sound and the manuscript well written, there are a number of points that I feel need to be clarified and/or adjusted. I present these as major, medium and minor points.

Major

1. Carriage surveys were conducted between 2016-2020, with PCV10 introduced in 2016. The major limitation I see in this manuscript is that the carriage reported for children in the <5yrs stratum includes both PCV-vaccinated and PCV-unvaccinated children. If true, the author needs to address this point in that the impact of vaccine-induced (direct) protection against carriage will be 'diluted' by including children not PCV-vaccinated. The author should consider reporting change in carriage prevalence over time among a younger subset of <5's that were PCV-vaccinated (e.g., children <2 years in 2017 vs children <2 years in 2019/2020).

The author often refers to determining "the population-level impact of PCV10 introduction on pneumococcal carriage and related this to PCV10 coverage in children aged <5 years." If the phrase "population-level impact of PCV10" implies a combined impact of direct and indirect protection, that should be clearly explained in the methods section, as well as presented as a Limitation.

2. The author needs to clearly address that, with each successive survey, a growing proportion of children in the <5yr stratum will have been PCV-vaccinated, increasing the contribution of indirect protection on controlling carriage. The author needs to address whether it's feasible to disentangle the combined contribution of direct and (increasing) indirect protection. If not feasible to disentangle, address what are the implications for interpreting these findings.

3. Line 128-132: It will be very informative for readers to have more information on these methodologies, including: i) Optichin sensitivity: Was any diameter of 'zone of inhibition' accepted as indicative of *Strep. pneumoniae* (Spn) or was there a minimum diameter. If there was a 'zone of inhibition' but less than the minimum acceptable, what was done (i.e., was bile solubility used?) ii) For serotyping, was a single colony picked and used to grow a pure-growth plate? iii) If yes, serotyping resulted in one serotype per plate? iv) Clarify how latex and Quellung complimented each another in this method to define serotype. v) Clarify how PCR and multiplex PCR complimented each another to define serotype in this method (e.g., if an isolate was positive by *lytA*, was this then serotyped using multiplex PCR?)

4. Line 144: How were 95% CI's calculated?

5. Figure 1 (Scatter diagram): It is not clear to me what is the denominator for 'PCV coverage'. In lines 155-156, the author writes: "We assessed PCV10 coverage...as the proportion of children who received two doses...". Is this coverage among those age-eligible for PCV at time of recruitment or among all children aged<5yrs irrespective of age-eligibility for PCV?

Medium

6. Abstract: It would help the reader to clarify in the abstract when PCV was introduced and using

what schedule.

7. Abstract: It would be more informative for the reader to report adjusted PRs in the abstract, rather than crude. If retain crude in the abstract, report them as 'Crude prevalence ratios...'

8. Line 60: Author writes "Between 2014 and 2016, Nigeria introduced..." Please add a sentence or two to briefly explain how PCV10 was rolled out (in a phased manner?) nationally.

9. Line 66: Author writes "Carriage studies conducted at the time of PCV10 introduction in Nigeria...". Clarify if these studies were done just before PCV introduction or just after PCV introduction. The term 'at the time' is somewhat ambiguous in this.

10. Line 67: Author writes "...showed overall carriage...". When using the term 'overall' for the first time in this context, please clarify what is meant. One option is "...showed overall (VT+NVT) carriage...". Also, the author uses different terms for 'overall.' As above, line 67 uses 'overall', Table 2 uses 'Any' and Line 139 refers to 'all pneumococci'. It is easier for the reader if the writing is consistent in use of such terminology.

11. Line 83: Author writes "The catchment areas...were those within 10km of ...Health Care Centres. Clarify why 10km was chosen.

12. Figure 1: I find the bar somewhat difficult to read, in knowing where 2016 ends and where 2017 starts...perhaps consider a different colour or shade for each year.

13. Line 97: Author writes "In the second stage of the carriage surveys..." This should probably be "In the second stage of sampling..."

14. Line 97. Sound like most of the methodology prior to this point in the text is about both carriage and coverage surveys. However, with line 97 and reference to choosing one participant per household, the author is referring only to the carriage survey. It would help the reader to ensure the methodology for carriage and coverage surveys are clearly differentiated when the method diverges.

15. Placement of Ref [13] at end of sentence suggests that it is a reference for the method of sample size calculation. I believe it is only a reference for the baseline 22-26%. One could write: "Sample size was informed by previous work...[13]"

16. Line 133-135: For the household visit for the coverage and carriage study, please clarify what was done if no one was home when the study team first visited the selected household. Was the next household on the list chosen, for example?

17. Line 133-135: Report what % of caregivers had the patient-retained 'vaccination cards' with the necessary vaccination history for the recruited child.

18. Line 139: Clarify what is 'LGA'

19. Line 147: The author refers to 'history of upper respiratory tract infection symptoms.' Clarify what symptoms were included. I suspect they are cough and runny nose, but it is important to make that link and to confirm for reader these were the only two symptoms.

20. Line 155-156: Author writes that they "assessed PCV10 coverage...as the proportion of children who received two doses of PCV10..." Below that (Line 162) and elsewhere the author refers to 3 doses. Is there a discrepancy in the text or does inclusion in different analyses require differing number of doses?

21. Figure 3 (Relationship between coverage): While the intention of the paper is not to compare rural to urban settings, clarify if the different time of recruitment (Dec in rural, Feb in urban) places them in different pneumococcal carriage seasons with different risks in carriage.

22. Figure 4 & 5: Consider providing prevalence data used to develop these figures in the Supplement. This would make it easier for the reader to evaluate what is written on line 224, for example: "The serotypes with the greatest reduction in carriage prevalence were 14, 19F, 23F and 6B."

23. Line 242 refers to "...53% and 64% cumulative vaccine coverage". Clarify at what time in this study period do these refer to...mid-point?

24. Line 282: Author writes: "...NVT replacement was limited to the population ≥ 5 years in the urban site." Table 2 suggests NVT replacement occurred in the rural settings (not urban) and likely for both < 5 (from 50-67%) and ≥ 5 (48-61%).

Minor

1. Abstract: Line 21: the word 'impact' is repeated twice.

2. Abstract: Line 24-25: Results (number of participants) is reported in the section that should be limited to 'Background'. I would recommend moving these down and clarifying what these numbers represent (i.e., total or 'per survey')

3. Line 52: Author writes "...from the vaccine-type (VT)..." I suggest "...from vaccine-serotype

(VT)..."

4. Author uses "vaccine-type", "vaccine-serotype" and sometimes "VT-serotype". It easier for the same term is used consistently.
5. Line 57: Author writes "...VT-serotype carriage and in IPD burden...". Should probably be "...VT carriage and in VT-IPD burden..."
6. Line 90-94: As written, sounds like the census was part of the 'first stage'...whereas I think the choosing of households from the census data was the first stage.
7. Line 110-111: Author writes "For the PCV10 coverage surveys, households were independently selected at random from the census described earlier." To what does the term independent refer? Perhaps write "...households were randomly selected..."
8. Line 122: Please report manufacturer of swab
9. Line 146-147: Author reports exposure variables included in the model. Consider reporting these only in the Results section.
10. Line 158: Author writes "...<5years...". Should be "...< 5years..." (add a space)
11. Table 1: Consider including a footnote to table that reminds reader that 'Clinical history' refers to 'reporting having experienced these symptoms in the two weeks preceding recruitment'
12. Table 1 & 2: The author refers to "under-fives" in these tables but "<5 years" in other places. Again, I think it's useful for the reader to be consistent.
13. Line 194: 069 should be 0.69?
14. Line 210: "Figure 1" should be "Figure 2"...ensure all the labelling and text are aligned. Be consistent in use of 'Fig' and 'Figure'.
15. Supplement: Table S1A: Should be changed to 'Landscape' layout or otherwise adjusted to fit the page.

Reviewer #2 (Remarks to the Author):

Adamu et al. present data on the impact of PCV10 on pneumococcal carriage in Nigeria, across various age strata, and sampling from an urban and rural location over 4-5 survey years. For most years, PCV10 coverage surveys were taken. Overall, the study is performed to a high standard, with overall appropriate methods and statistical analysis. It is well written and an interesting read. There are some inconsistencies, for example the missing vaccine coverage surveys early in the study. These reflect the real-world complexities of working in low-middle income settings, where the disease burden is greatest and the data most valuable. The study is from an important and populous country in Africa.

Specific comments and issues are detailed below:

Major issues

1. Serotype replacement
 - a. I think that the replacement (particularly in rural >5years) is a major finding here but given insufficient attention in the manuscript, (including but not limited to the abstract)
 - b. In the results I wanted to understand which NVT had arisen. Perhaps a line graph similar to SuppF2 in Dunne et al. Lancet Global Health 2018 may be informative (together, or with VT and NVT as separate panels)?
 - c. How do the replacing serotypes relate to PCVs on or approaching the market? How does this study inform vaccine policy in Nigeria?
 - d. Serotype 3 (also see comments in microbiology) is unlikely to be an innocuous replacer, given this has emerged as a key IPD serotype post-PCV in other settings. This should be highlighted.
2. Association of coverage and VT prevalence. I think this area can be strengthened.
 - a. I was not overly convinced on the assertion that these relationships are linear, e.g. for the urban under 5s. Would other relationships better explain the data? Should there be error bars? Are there enough datapoints for this analysis, what is the uncertainty?
 - b. How do these analyses compare with other studies such as Chan et al. Lancet Regional Health Western Pacific 2021?
 - c. A useful output from this analysis is to understand the coverage required to get X% of VT reduction, how do these results compare with other settings?
3. Sampling strategy, was the strategy to enrol 100/group? Why were some age strata so over enrolled? Also seemed to vary by survey.
4. What was the age range in the ≥ 40 s, can this be appropriately presented as a single age

group?

5. Urban vs rural

a. I think there is perhaps two ways to approach presentation of data from the two sites. One is to present them as individual sites (location A and B) and then present some reasons for the differences between sites (rural/urban). The other is to present them as exemplar rural and urban sites, fully contrasting and comparing between them. In the latter, the only differences between surveys should be 'urban-ness'. I felt the paper switched back and forth between these two approaches.

b. Related to above, can urban and rural be reasonably compared e.g. in a supplemental analyses? If so, it would be nice to have a table on this. Also may give capacity to draw out some of the other observations, such as although there was higher carriage prevalence in rural settings prior to PCV, what was the diversity and how does this relate to the observations on PCV impact and replacement?

c. Surveys were only a few months apart, but does seasonality play a role here in being able to directly compare these locations?

d. Serotypes for urban and rural are presented in Fig 4, but not much is made of the differences/similarities in the discussion. Perhaps this is more a supplement figure, with the main figure being serotype by <5 and >5 from both sites?

e. What strategies were employed (consultation periods, employment of local staff) to ensure best recruitment and appropriateness of the rural survey. How does the low recruitment compare with other studies in this region?

6. Microbiology

a. How were colonies selected for analysis? Was it one colony of the major type, a randomly selected colony, one of each morphology? I presume multiple carriage would be quite high in this setting and so the microbial sampling strategy could have a big effect on serotype data.

b. Related to above, the colony sampling strategy may have affected results such as the high prevalence of serotype 3 (which looks different on the plate) and some of the assertions around potential for unmasking etc. Would it be valuable to do a subset by microarray, or limit analysis to the dominant colony only?

c. In what way were Quellung results inconclusive? Was this because of an incomplete set of antisera, difficulty in resolving factors, or some other issue? It is difficult to understand whether these issues are methodological, or may for example reflect true diversity (such as novel serotype variants or new serotypes that might be real results and hidden by incorrect molecular serotyping). Perhaps the small number of isolates can be retested by an external laboratory? Or tested with whole-genome sequencing?

d. Serotype 3 a big issue (particularly in rural) and was common even prior to vaccine introduction. How were these typed (also relates to point 5b) and where these arising from clonal expansion?

7. Perhaps I'm missing something, but why was the vaccine coverage of individual study participants not gathered?

8. Table 1, antibiotic use seemed to vary a lot, was this more than expected? were there circulating viruses/outbreaks at time that also may affect your analysis?

9. Was a birth cohort analysis also done in later years that can be included to provide assurance similar results to the PCV10 survey?

10. How was the 'reduction in carriage prevalence' assessed, for example if reduced from 4% to 2% is this a 50% reduction or a 2% reduction (vs change from 40% to 20%, is this 50% or 20%)? Please edit to clarify.

11. There are not many studies of PCV impact in LMICs. Suggest comparing your findings against them all in the discussion to provide a complete picture, including for vaccination coverage.

Minor issues

1. Line numbers would be helpful

2. I don't think SRD is an informative acronym

3. I didn't find Figure 1 particularly informative, perhaps this timeline could be in the supplement, leaving another figure for data or a participant flow chart?

4. Italicise gene names (lytA)

5. Tables, I think additional explanatory footnotes would be helpful to aid the reader (without having to consult text)

6. Related to above, for Table 1 specify whether current (or within timeframe X) clinical

- symptoms, whether household composition includes the participant
7. Table 1, was only data on solid fuel collected?
 8. Colours in Fig 4 are not informative, suggest colour VT and NVTs. If use colour, then each serotype should be a different colour so the reader can easily see changes over time.
 9. Did I miss the serotype data for the <5s?
 10. Check wording in Discussion "However, NVT replacement was limited to the population ≥ 5 years in the urban site." (limited in urban, limited to rural?)
 11. Is there evidence that replacement serotypes were present at low abundance (e.g. by microarray)? Given that unmasking does not appear to be a major contributor in the literature, and that the relationship of PCV and density is very unclear that this stage, suggest change "likely possibility" to "possibility" when talking about unmasking/density.
 12. References – italicise species (and gene names if present)

Reviewer #3 (Remarks to the Author):

This manuscript reflects important work carried out in a country with a very high burden of pneumococcal and pneumococcal related disease. Thus the need for effective vaccines and vaccine programs is particularly important and the authors' work is informing these issues.

The manuscript, as written, does not provide sufficient detail about the study methods and there are numerous errors in the labeling of tables and figures, some of which are missing from the material that I reviewed. The materials that I reviewed included the manuscript (PDF and Word versions, which contained text, Figures 1 – 5 and Tables 1-3), a supplement (PDF and Word, which contained Tables S1A & S1B, and Figures S2A and S2B), a related manuscript (PDF) and a data set (Excel).

The comments that follow are provided in the order that each issue was encountered in the manuscript.

Page 5/Line 85ff, Figure 1 and Results. The authors state that "baseline carriage surveys were conducted" and published (reference #13, which they provided) but they do not clearly state that the data from this prior publication was included in this current manuscript. This should explicitly state and the authors should describe if there is additional or modified data from that prior publication in the current manuscript.

Page 5/Line 85ff and Discussion. The authors should acknowledge that not having pre-vaccine program baseline carriage data is a limitation as discussed in reference #1 (Flasche et al).

Page 5/Line 90ff. The authors should provide a rationale for their sampling design. In addition, it seems that it is really a three-stage design since there was already a selection of one rural community and one urban community, before the next stages of household census and selection and age-stratified selection from within households.

Page 7/Line 98ff, Tables 1, S1A,S1B. The authors state that they selected participants from 10 age groups but they reported their age-specific data using just 5 age groups which contained variable proportions of the pre-defined age groups (<5 (3 age groups), 5-9 (1 age group), 10-19 (2 age groups), 20-39 (1 age group) ≥ 40 (3 age groups)). The actual proportions of participants from these combined age groups do not reflect equal proportions from each of the 10 pre-defined age groups. The authors should better explain their recruitment by age group sequence and how they managed missing age groups from within household, which appears to be mainly an issue with lower proportions of adult participants.

Page 7/Line 104ff and Line 113ff and Discussion. The authors provide two different sample size estimates, one based on detection of reduction of vaccine serotype carriage and one based on estimated of vaccine coverage. This seems superfluous. The authors did not state what their primary outcome of interest was and so it is not clear which sample size estimate was more important. Regardless, they did not meet their sample size goal by either estimate for any of the

surveys (100 participants in each age group in each survey or 639 children in each survey). This should be acknowledged in the discussion of limitations.

Page 10/Line 170ff. The authors should state whether there were participants who took part in two or more surveys. They should also state whether there were two or more participants from individual households in any of the surveys – presumably not based on their description of the recruitment, but this should be stated explicitly.

Page 10/Line 173. The authors refer to Table S1. Do they mean Table S1A and S1B? These distinct tables should be referred to separately. Also there is no Figure S1. Do they mean Figure S2A and Figure S2B?

Page 12/Line 180. There is no Figure S3.

Page 12/Line 181ff, Table 2 and Page 15/Line 192ff and Table 3. There was a statistically significant reduction in overall pneumococcal carriage in children under 5 years of age in the urban site but not the rural site. The authors state this but do not reflect on it in the discussion. Declines in overall carriage after introduction of pneumococcal conjugate vaccine programs have been reported in other countries, particularly high income countries where the force of infection is lower (and often much lower) than in low and middle income countries. Do the authors think that there was a significantly lower force of infection in their urban setting vs their rural setting?

Page 20/Figure 3. The scale or percentages on the y-axis of each of the figures should be the same. As presented the y-axis on the left goes from 0-80% and on the right from 0-100%.

Page 20/Line 225. There is not Table S2.

Page 20/Lines 225ff, Figure 4 and Figure 5. Both Figure 4 and Figure 4 are labelled as referring to persons ≥ 5 years old in the rural sites, but the text refers to all ages and both the rural and urban sites. This is confusing and should be clarified or corrected.

Reviewer #1

Major

1. Carriage surveys were conducted between 2016-2020, with PCV10 introduced in 2016. The major limitation I see in this manuscript is that the carriage reported for children in the <5yrs stratum includes both PCV-vaccinated and PCV-unvaccinated children. If true, the author needs to address this point in that the impact of vaccine-induced (direct) protection against carriage will be 'diluted' by including children not PCV-vaccinated. The author should consider reporting change in carriage prevalence over time among a younger subset of <5's that were PCV-vaccinated (e.g., children <2 years in 2017 vs children <2 years in 2019/2020).

Response:

- This response indicated to us that we had failed to communicate the main purpose of the study, which is to look at the impact of PCV introduction into the routine immunisation programme on carriage at a population level. This population-level impact would include both the direct effect, among vaccinated children, and the indirect effect among the whole population, attributable to reduced transmission. The design of the study would not allow us to tease out direct from indirect effects among the vaccine-eligible children, nor was that our purpose. Our aim was to demonstrate the impact of the vaccine programme in the overall population because this type of evidence of public health impact is more likely to be effective in shaping policy. Based on this and similar comments from other reviewers, we accept that we may not have conveyed our aim clearly. We have now re-written the introduction and substantial sections of the discussion to clarify these aims and thank the reviewer for his/her frank comments.
- We used age groups <5 and ≥5 years because children aged <5 years bear the largest burden of carriage, are the main reservoirs for transmission and have the highest risk of pneumococcal disease. Indirect effects of the programme are,

therefore, likely to be strongly correlated with total vaccine coverage among children aged <5 years. A study of narrower age brackets might attempt to examine the contribution of direct and indirect effects but this would really require knowledge of carriage and vaccine coverage in the same individuals whilst in this study, these variables are estimated at population level in different samples of the population.

- 2. The author often refers to determining “the population-level impact of PCV10 introduction on pneumococcal carriage and related this to PCV10 coverage in children aged <5 years.” If the phrase “population-level impact of PCV10” implies a combined impact of direct and indirect protection, that should be clearly explained in the methods section, as well as presented as a Limitation.**

Response:

- We agree with the reviewer that the population-level impact of PCV10 comprises a combination of direct protection of vaccinated children and indirect protection to unvaccinated children and adults. We designed this study to capture the impact of the vaccine on nasopharyngeal carriage which includes a combination of direct and indirect vaccine-induced effects. We did not aim to tease out direct from indirect effects and we do not see this as a limitation. Our approach allows us to assess the impact on carriage of the vaccination coverage among children aged <5 years who are also the main reservoirs for transmission. We have re-written the introduction and sections of the discussion to try to describe our goal more clearly.

- 3. The author needs to clearly address that, with each successive survey, a growing proportion of children in the <5yr stratum will have been PCV-vaccinated, increasing the contribution of indirect protection on controlling carriage. The author needs to address whether it’s feasible to disentangle the combined contribution of direct and (increasing) indirect protection. If not feasible to disentangle, address what are the implications for interpreting these findings.**

Response:

- We agree with the reviewer that expanding proportions of PCV-vaccinated children in successive surveys would also increase the indirect protection against carriage. This is exactly what happens when PCV10 is introduced into childhood immunisation schedules without a catch-up campaign.
- However, as stated above, our aim was to determine these combined effects on carriage in the whole population and not to quantify direct vs indirect effects of vaccination.

4. Line 128-132: It will be very informative for readers to have more information on these methodologies, including: i) Optichin sensitivity: Was any diameter of 'zone of inhibition' accepted as indicative of Strep. Pneumoniae (Spn) or was there a minimum diameter. If there was a 'zone of inhibition' but less than the minimum acceptable, what was done (i.e., was bile solubility used?) ii) For serotyping, was a single colony picked and used to grow a pure-growth plate? Iii) If yes, serotyping resulted in one serotype per plate? Iv) Clarify how latex and Quellung complimented each another in this method to define serotype. V) Clarify how PCR and multiplex PCR complimented each another to define serotype in this method (e.g., if an isolate was positive by lytA, was this then serotyped using multiplex PCR?)

Response:

- We have added more information Pg#19 , lines 334-340-
"We identified pneumococci by alpha-haemolysis and optochin sensitivity testing. For optochin-resistant isolates (zone of inhibition <14mm diameter), we used bile solubility testing to confirm S. pneumoniae. For serotyping, we selected one colony per plate from the dominant colony morphology. We identified serotypes using latex agglutination confirmed by Quellung Reaction. For isolates with inconclusive serotyping, we confirmed species and serotype by polymerase chain reaction (PCR) for autolysin (lytA) and capsular locus genes, respectively."

5. Line 144: How were 95% CI's calculated?

Response:

- 95% CIs were from the stata output for respective regression models (log-binomial or Poisson regression). Using the log link function and appropriate family (binomial or Poisson), Stata by default reports risk ratios (exponentiated regression co-efficients) estimated by the respective models with the 95% CIs of these ratios.

6. Figure 1 (Scatter diagram): It is not clear to me what is the denominator for 'PCV coverage'. In lines 155-156, the author writes: "We assessed PCV10 coverage...as the proportion of children who received two doses...". Is this coverage among those age-eligible for PCV at time of recruitment or among all children aged<5yrs irrespective of age-eligibility for PCV?

Response:

- This has now been clarified in the text of the methods section Pg #21, lines 367-370. *"The purpose of the coverage survey was to infer population immunity, not to evaluate programme effectiveness. Therefore, we estimated PCV10 coverage in each survey year (2018-2020) as the proportion (with 95%CI) of children aged <5 years (regardless of age-eligibility) who received two doses of PCV10 irrespective of timing and age of receipt.."*

Medium

6. Abstract: It would help the reader to clarify in the abstract when PCV was introduced and using what schedule.

Response:

- We have now indicated the dates of PCV introduction in both sites and the schedule in the Introduction section Pg #4, lines 40-42.

- We have also added the year of PCV introduction in these sites in the abstract Pg #2, line 29.

7. Abstract: It would be more informative for the reader to report adjusted PRs in the abstract, rather than crude. If retain crude in the abstract, report them as ‘Crude prevalence ratios...’

Response

- As the abstract is required to be shorter than in our submission, and as the prevalence ratios are not immediately accessible as a concept, we have simply reported the prevalence results (baseline and final survey) in the abstract. In the results section we have limited the analyses to PRs comparing carriage in the final surveys to the baseline surveys for each site and focused the reporting only on adjusted age-standardized PRs.

8. Line 60: Author writes “Between 2014 and 2016, Nigeria introduced...” Please add a sentence or two to briefly explain how PCV10 was rolled out (in a phased manner?) nationally.

Response:

- We have now added a brief description of the PCV roll-out in Nigeria in Pg #4, lines 40-42 –

“Between 2014 and 2016, in three geographically distinct phases, Nigeria introduced the 10-valent Pneumococcal Conjugate Vaccine (PCV10) in a three-dose schedule for infants aged 6, 10 and 14 weeks, without a catch-up campaign”

9. Line 66: Author writes “Carriage studies conducted at the time of PCV10 introduction in Nigeria...”. Clarify if these studies were done just before PCV introduction of just after PCV introduction. The term ‘at the time’ is somewhat ambiguous in this.

Response:

- We have clarified that the studies were done just after PCV introduction. Pg #5, lines 70-72. The issue is also detailed clearly in Figure 1.

“In Nigeria, among children aged <5 years who were studied immediately after PCV10 introduction...”

10. Line 67: Author writes “...showed overall carriage...”. When using the term ‘overall’ for the first time in this context, please clarify what is meant. One option is “...showed overall (VT+NVT) carriage...” Also, the author uses different terms for ‘overall.’ As above, line 67 uses ‘overall’, Table 2 uses ‘Any’ and Line 139 refers to ‘all pneumococci’. It is easier for the reader if the writing is consistent in use of such terminology.

Response:

- We have clarified (page #5, lines 75-76) overall carriage to refer to any pneumococcal carriage and (p5, line 97) total population carriage to refer to carriage in all ages and made this consistent throughout the text.

11. Line 83: Author writes “The catchment areas...were those within 10km of ...Health Care Centres. Clarify why 10km was chosen.

Response:

- We chose 10km as advised by our local collaborators because this is the catchment area for the respective health care centres. This also facilitated access and community entry, as residents of catchment communities were familiar with the health care workers.

12. Figure 1: I find the bar somewhat difficult to read, in knowing where 2016 ends and where 2017 starts...perhaps consider a different colour or shade for each year.

Response:

- We have alternated the colours for the different years

13. Line 97: Author writes “In the second stage of the carriage surveys...” This should probably be “In the second stage of sampling...”

Response:

- We have revised the statement in Pg #17, line 298 to: *“In the second stage of sampling....”*

14. Line 97. Sound like most of the methodology prior to this point in the text is about both carriage and coverage surveys. However, with line 97 and reference to choosing one participant per household, the author is referring only to the carriage survey. It would help the reader to ensure the methodology for carriage and coverage surveys are clearly differentiated when the method diverges.

Response:

- We have edited the text to support this distinction. At the top of page 17 we have described the processes in common, ending with *“We selected separate samples of households for the carriage and PCV10 coverage surveys”*. Then we have described the second stage of sampling separately for the two types of survey in quite separate paragraphs. *“In the second stage of sampling for the carriage survey”* (p17, line 298) and *“In the second stage of sampling for the PCV10 coverage survey”* (p18, line 315)

15. Placement of Ref [13] at end of sentence suggests that it is a reference for the method of sample size calculation. I believe it is only a reference for the baseline 22-26%. One could write: “Sample size was informed by previous work...[13]”

Response:

- We have revised the statement in page #17, lines 310-311, as follows:
“For the baseline carriage surveys (2016)[12] the sample size was set at 1000 participants to achieve a desired precision.”

16. Line 133-135: For the household visit for the coverage and carriage study, please clarify what was done if no one was home when the study team first visited the selected household. Was the next household on the list chosen, for example?

Response:

- If the household was known to be occupied, but there was no one at home, we revisited later. If the house was non-residential, unoccupied or empty, we chose the next household on the list.
- We have also included this in the methods section, page #17, lines 295-297

17. Line 133-135: Report what % of caregivers had the patient-retained ‘vaccination cards’ with the necessary vaccination history for the recruited child.

Response:

- We have provided more information about the card retention in the Result section pages #8-9, lines 138-141 as reproduced below.

“The average proportion of children for whom the caregivers had retained their vaccination card was 70% in the rural site (52% in 2018; 77% in 2019; and 90% in 2020) and 80% in the urban site (70% in 2019 and 91% in 2020).”

18. Line 139: Clarify what is ‘LGA’

Response:

- We have now written it full on page #19, lines 353: “...Local Government Areas (LGAs).”

19. Line 147: The author refers to ‘history of upper respiratory tract infection symptoms.’ Clarify what symptoms were included. I suspect they are cough and runny nose, but it is important to make that link and to confirm for reader these were the only two symptoms.

Response:

- The upper respiratory tract infection symptoms are now clearly stated as cough and runny nose in page #20, lines 361.

20. Line 155-156: Author writes that they “assessed PCV10 coverage...as the proportion of children who received two doses of PCV10...” Below that (Line 162) and elsewhere the author refers to 3 doses. Is there a discrepancy in the text or does inclusion in different analyses require differing number of doses?

Response:

- We accept this was inconsistent. We have now used receipt of two doses of PCV10 consistently in all analyses as the relevant marker of coverage.

21. Figure 3 (Relationship between coverage): While the intention of the paper is not to compare rural to urban settings, clarify if the different time of recruitment (Dec in rural, Feb in urban) places them in different pneumococcal carriage seasons with different risks in carriage.

Response:

- The two sites also have different seasons. Therefore, we cannot control for both site and season at the same time. However, within each site, we deliberately conducted the carriage surveys at the same period for each year to avoid differential seasonal effects. As understood, the selection of rural and urban sites was purposefully done to illustrate epidemiological differences rather than to specify the exact variation in carriage prevalence by site; it would require a substantial increase in word count to try to characterise the potential impact of different months of sampling and the data (ie VT carriage prevalence throughout the seasons) is not available to inform this potential discussion.

22. Figure 4 & 5: Consider providing prevalence data used to develop these figures in the Supplement. This would make it easier for the reader to evaluate what is written on

line 224, for example: “The serotypes with the greatest reduction in carriage prevalence were 14, 19F, 23F and 6B.”

Response:

- We had already included the carriage prevalence of all serotypes (stratified as children aged <5 years and persons aged ≥5 years) identified in all the surveys in the supplement (these are now Tables S2-S3). We had also provided the numbers of carriage isolates for all serotypes identified in all the surveys, stratified the age groups as Supplement data.

23. Line 242 refers to “...53% and 64% cumulative vaccine coverage”. Clarify at what time in this study period do these refer to...mid-point?

Response:

- We have now re-written this section and deleted this figure. We have now consistently referred to annual PCV10 coverage of ≥2 doses.

24. Line 282: Author writes: “...NVT replacement was limited to the population ≥5 years in the urban site.” Table 2 suggests NVT replacement occurred in the rural settings (not urban) and likely for both <5 (from 50-67%) and ≥5 (48-61%).

Response:

- We have re-written it in a clearer way in page #11, lines 189-191 –
“The decline in VT carriage prevalence in Nigeria was accompanied by an increase in NVT carriage prevalence among children in Kumbotso (rural) and among older persons in both settings, with adjusted prevalence ratios of 1.26-1.34.”

Minor

25. Abstract: Line 21: the word 'impact' is repeated twice.

Response:

- Thank you for pointing this out. In light of the issues raised by all reviewers, we have now re-written the abstract.

26. Abstract: Line 24-25: Results (number of participants) is reported in the section that should be limited to 'Background'. I would recommend moving these down and clarifying what these numbers represent (i.e., total or 'per survey')

Response:

- We have taken this into consideration in the now re-written abstract.

27. Line 52: Author writes "...from the vaccine-type (VT)..." I suggest "...from vaccine-serotype (VT)..."

Response:

- This has been revised as suggested.

28. Author uses "vaccine-type", "vaccine-serotype" and sometimes "VT-serotype". It easier for the same term is used consistently.

Response:

- We have now restricted to VT as acronym for vaccine-serotype and made it consistent throughout the text.

29. Line 57: Author writes "...VT-serotype carriage and in IPD burden...". Should probably be "...VT carriage and in VT-IPD burden..."

Response:

- We have re-written this section and have now deleted the phrase. But elsewhere in the discussion section we have written this as VT IPD consistently.

30. Line 90-94: As written, sounds like the census was part of the ‘first stage’ ...whereas I think the choosing of households from the census data was the first stage.

Response:

- Although the census was a key step to selecting the households, because there was no existing sampling frame, the first stage of sampling was the selection of households from the census listing. We have re-written this sub-section to clarify in page #17, lines 291-294, reproduced below:

“Having selected representative study areas, we used a two-stage sampling design. In the first stage, we selected households using simple random sampling. To obtain a sampling frame, we conducted a census of all households in the catchment area before each survey.”

31. Line 110-111: Author writes “For the PCV10 coverage surveys, households were independently selected at random from the census described earlier.” To what does the term independent refer? Perhaps write “...households were randomly selected...”

Response:

- Although households were selected from the same census list for both carriage and vaccine coverage surveys, we selected households for the PCV10 coverage surveys independent from the households selected for the carriage survey.
- We have clarified this in the revised text on pages page #17, lines 294-295.

“We selected separate samples of households for the carriage and PCV10 coverage surveys.”

32. Line 122: Please report manufacturer of swab

Response:

- Done

33. Line 146-147: Author reports exposure variables included in the model. Consider reporting these only in the Results section.

Response:

We agree with the reviewer these variables are analytic outputs. For ease of communication, we have opted to report the variables included in the models in the methods section.

34. Line 158: Author writes "...<5years...". Should be "...< 5years..." (add a space)

Response:

We have included a space, though between <5 and years. Inspection of a number of prior publications in Nature Communications suggests acceptable precedents for both "< 5 years" and "<5 years" and we have adopted the latter consistently throughout.

35. Table 1: Consider including a footnote to table that reminds reader that 'Clinical history' refers to 'reporting having experienced these symptoms in the two weeks preceding recruitment'

Response

Thank you. We have included this in the footnote.

36. Table 1 & 2: The author refers to "under-fives" in these tables but "<5 years" in other places. Again, I think it's useful for the reader to be consistent.

Response:

We have adopted “aged <5 years” consistently throughout the text.

37. Line 194: 069 should be 0.69?

Response:

Thank you. We have corrected this.

**38. Line 210: “Figure 1” should be “Figure 2”...ensure all the labelling and text are aligned.
Be consistent in use of ‘Fig’ and ‘Figure’**

Response:

This has been done. We have adopted “Figure” consistently throughout.

**39. Supplement: Table S1A: Should be changed to ‘Landscape’ layout or otherwise
adjusted to fit the page.**

Thank you for pointing this out. We have now merged Tables S1A and S1B (now Table S1) and formatted the layout of the new table appropriately.

Reviewer #2

Major issues

1. Serotype replacement

- a. **I think that the replacement (particularly in rural >5years) is a major finding here but given insufficient attention in the manuscript, (including but not limited to the abstract)**
- b. **In the results I wanted to understand which NVT had arisen. Perhaps a line graph similar to SuppF2 in Dunne et al. Lancet Global Health 2018 may be informative (together, or with VT and NVT as separate panels)?**
- c. **How do the replacing serotypes relate to PCVs on or approaching the market? How does this study inform vaccine policy in Nigeria?**
- d. **Serotype 3 (also see comments in microbiology) is unlikely to be an innocuous replacer, given this has emerged as a key IPD serotype post-PCV in other settings. This should be highlighted.**

Response:

- a. Serotype replacement carriage is certainly an issue to watch. The prevalence ratios we observed were relatively small compared to elsewhere in Africa and we have discussed this in the discussion as a key issue. Higher relative increases in serotype replacement carriage have not translated to serotype replacement disease in Africa (e.g. Kenya) but the data from across the continent are too scanty to draw firm conclusions. We have reduced the length of the abstract substantially to comply with Nat Comms specification and this has meant cutting out elements – it is difficult to add in new elements and we have focused on the impact on VT serotypes which was the primary objective of the paper
- b. We have now included graphs to illustrate the distribution and ranking of serotypes in in the baseline and final carriage surveys by age group and site with colour codes to indicate VTs and NVTs (Figure 2). We have also included tables to show annual serotype-specific carriage prevalence of serotypes and 95%

confidence intervals by age and site (Tables S2-S3). The tables illustrate how little precision the study has in defining the prevalence of individual serotypes and, consequently, we have not emphasised the comparison of serotypes from survey to survey.

- c. We have made a comparison of carriage prevalence of serotypes in the different licensed PCV formulations in the supplement (Table S5). At present the study provides information to policy makers only on the impact of the vaccine on vaccine serotypes (its intended target). The impact of serotype replacement carriage cannot yet be assessed until there are data from elsewhere in Africa on carriage and disease where the invasive potential of NVTs can be assessed.
- d. Carriage prevalence of serotype 3 did not change significantly in either site, as shown in Tables S2-S3.

2. Association of coverage and VT prevalence. I think this area can be strengthened.

- a. **I was not overly convinced on the assertion that these relationships are linear, e.g. for the urban under 5s. Would other relationships better explain the data? Should there be error bars? Are there enough datapoints for this analysis, what is the uncertainty?**
- b. **How do these analyses compare with other studies such as Chan et al. Lancet Regional Health Western Pacific 2021?**
- c. **A useful output from this analysis is to understand the coverage required to get X% of VT reduction, how do these results compare with other settings?**

Response:

- We thank the reviewer for suggesting the article. Our dataset for the vaccine coverage is at a population level and not individual-level, so we have very few datapoints. We were only able to conduct a simple ecological analysis to visualise the relationship between VT carriage and vaccine coverage. For consistency with other analyses (Reviewer 1, point 20) we have re-analysed this relationship using ≥ 2 doses as the coverage marker (previously it was 3 doses) and the new graph suggests a non-linear relationship, at least for children < 5 years. Theoretical considerations also suggest the relationship is non-linear, and varying with time, and we have now

emphasised in the discussion how this may be better captured through transmission dynamic modelling (which is beyond the scope of this paper). For this reason, we have avoided translating the ecological regression into quantitative predictions as proposed (2c). In contrast to these considerations, the data for older persons does now appear to follow a linear relationship within the range of childhood vaccine coverage levels investigated. Whilst the development of 'herd protection' is undoubtedly more complex, the linear reduction in VT carriage among older children/adults with rising PCV10 uptake does, at a minimum, provide local policy makers with evidence that the vaccine programme is having an indirect effect in this age-group and that improvements in coverage, in areas that have lagged behind, may provide relatively predictable gains in reducing transmission of VT pneumococci.

3. Sampling strategy, was the strategy to enrol 100/group? Why were some age strata so over enrolled? Also seemed to vary by survey.

Response:

- Yes, we had an age-stratified sampling procedure to ensure we recruited across all ages in the population with a target to recruit 100/age group to make up a minimum sample of 1000 per survey
- Here we offer some reasons for the variation in participation rates. We have also indicated this in the footnote for Table S1.
 - Some age groups declined to be swabbed after agreeing to participate, with the main reason being believing that the procedure was too 'invasive'. This was particularly common among school-aged children and young adolescents. To overcome this anticipated limitation, we oversampled such age groups.
 - There were sometimes errors in how the ages were reported at recruitment. When this occurred, we reclassified participants into the correct/appropriate age groups at the time of interview. In such cases, we resampled from the wrongfully assigned age group and this may have influenced the numbers of participants that were eventually swabbed in each age group.

- The ethos of the study was to find a simple and inexpensive way of providing policy relevant information on the impact of PCV10. The funding for fieldwork was very modest and insufficient to correct all potential biases arising during sampling.

4. What was the age range in the ≥ 40 s, can this be appropriately presented as a single age group?

Response:

- Age in years, median (IQR) and range
 - Rural: Median 54 years, IQR 45-65 and range 40-100 years
 - Urban: Median 52 years, IQR 46-61 and range 40-101 years
- The key consideration here is the proportion of the population that falls within this age-group. Proportion of residents aged ≥ 40 years were 19% in Kumbotso and 28% in Pakoto. These age groups constituted 12% of overall carriers and 7% of VT carriers in Kumbotso and 15% and 11% in Pakoto, respectively. For this reason we did not subdivide this age stratum into finer classes.

5. Urban vs rural

- a. I think there is perhaps two ways to approach presentation of data from the two sites. One is to present them as individual sites (location A and B) and then present some reasons for the differences between sites (rural/urban). The other is to present them as exemplar rural and urban sites, fully contrasting and comparing between them. In the latter, the only differences between surveys should be 'urban-ness'. I felt the paper switched back and forth between these two approaches.**
- b. Related to above, can urban and rural be reasonably compared e.g. in a supplemental analyses? If so, it would be nice to have a table on this. Also may give capacity to draw out some of the other observations, such as although there was higher carriage prevalence in rural settings prior to PCV, what was the diversity and how does this relate to the observations on PCV impact and replacement?**

- c. **Surveys were only a few months apart, but does seasonality play a role here in being able to directly compare these locations?**
- d. **Serotypes for urban and rural are presented in Fig 4, but not much is made of the differences/similarities in the discussion. Perhaps this is more a supplement figure, with the main figure being serotype by <5 and >5 from both sites?**
- e. **What strategies were employed (consultation periods, employment of local staff) to ensure best recruitment and appropriateness of the rural survey. How does the low recruitment compare with other studies in this region?**

Response:

- a. Our intention was to select two sites that were exemplars of rural and urban environments, as we anticipated the epidemiology of *S. pneumoniae* would differ with urban/rural status. To clarify this approach we have made it explicit at the beginning of the Methods section and have subsequently referred to the two sites in the text as 'rural' or 'urban', and in the tables and figures as 'Kumbotso (rural)' and 'Pakoto (urban)'.
- b. We have included a table in the supplement showing the number of serotypes and diversity indices by site, over time in the two age groups.
- c. Although, we deliberately conducted the carriage surveys at the same period for each year to avoid differential seasonal effects within site, from year to year, they both experience varying effects of the harmattan season between November and March according to proximity to the savannah belt and Sahara Desert.
The harmattan season peaks in December in the rural northern site where it is dry and dusty and the temperature drops considerably, and in the south, it is also dry and dusty, but the temperature and humidity do not drop as much.
Therefore, we can not separate the effect of season and site at the same time.
- d. Comparison of the serotypes in the two sites was one of the objectives in the baseline survey, published previously (ref 12). The objective of the current study is to examine the impact of PCV10 introduction on carriage and the sample size was set to test this question for groups of serotypes (e.g. VT and NVT) so we have

relegated the impact on individual serotypes to supplementary tables 2/3. Analysis of the two sites illustrates that there has been an impact in each; we did not set the sample size to study an interaction by rural/urban setting and so we have not tested this hypothesis. The difference in serotype distributions at these two sites at different time points is not the focus of the study and this explains why it is given relatively little highlight in the discussion. We are interested in serotype replacement carriage (analysed as NVT) but, again, the power to examine individual serotypes is limited. Nonetheless, we have provided data for those who are interested in this question in Figure 2 and in supplementary tables 2/3 and highlighted in the results text the few serotypes that showed significant changes over time, in each site.

e. To ensure good participation and cooperation:

- We worked together with staff from the respective health centres that have been conducting community work and providing health services to the communities
- We obtained permission from the respective LGA departments of health following IRB approvals
- Before each survey, we made advocacy visits to the traditional leaders and health facility development committees and religious leaders
- In the rural site, via the traditional district, ward and village heads we organised for daily town announcers to communicate to residents on daily basis throughout the survey period. We also worked with volunteer communication mobilizers (VCMs) to conduct census, recruit, and invite and send reminders to participants. In the urban site, we worked with community representatives nominated by the community leader.

Annual recruitment ranged between 88-99% in the rural site and between 85-94% in the urban site. Given the target of 1000 participants per survey, we had a good participation for the carriage survey. Carriage surveys in other African countries reported participation levels of 71-78% in Kilifi, Kenya, 92-96% in Malawi, 81-94% in Gambia, and 71% in Cameroon.

6. Microbiology

- a. **How were colonies selected for analysis? Was it one colony of the major type, a randomly selected colony, one of each morphology? I presume multiple carriage would be quite high in this setting and so the microbial sampling strategy could have a big effect on serotype data.**
- b. **Related to above, the colony sampling strategy may have affected results such as the high prevalence of serotype 3 (which looks different on the plate) and some of the assertions around potential for unmasking etc. Would it be valuable to do a subset by microarray, or limit analysis to the dominant colony only?**
- c. **In what way were Quellung results inconclusive? Was this because of an incomplete set of antisera, difficulty in resolving factors, or some other issue? It is difficult to understand whether these issues are methodological, or may for example reflect true diversity (such as novel serotype variants or new serotypes that might be real results and hidden by incorrect molecular serotyping). Perhaps the small number of isolates can be retested by an external laboratory? Or tested with whole-genome sequencing?**
- d. **Serotype 3 a big issue (particularly in rural) and was common even prior to vaccine introduction. How were these typed (also relates to point 5b) and where these arising from clonal expansion?**

Response:

- a. We selected one colony per plate based on morphology. If more than one morphology was identifiable, we selected one colony from each morphology. We included only the dominant colony in this analysis., i.e., one isolate per swab/participant. Although this method could underestimate serotype-specific carriage and bias towards overestimating morphologically distinct serotypes, it would give a general picture of the relative prevalence of circulating serotypes.
- b. We agree that the unique morphology of serotype 3 could be a source of bias, however, we analysed the dominant morphotype consistently throughout the study and the carriage prevalence for serotype 3 did not change significantly (See suppl Tables S2-S3).

- c. Inconclusive Quellung refers to isolates for which serotypes could not be assigned, i.e., negative with all the antisera pools and/or negative to the Omniserum reagent. For these isolates, we used polymerase chain reaction (PCR) targeting the genes encoding autolysin (*lytA*) was used to confirm *S. pneumoniae* and then multiplex PCR was used to identify the serotype. We have clarified the laboratory procedure in page #19, lines 334-340.
- d. Serotype 3 is consistently among the seven most common NVT in older children and adults, though it is much less prominent in children (who are the primary focus of this study). As noted above, the prevalence of serotype 3 did not change significantly in either age group or in either site, with the introduction of PCV10. Of note, the study was not sample-sized to draw conclusions about individual serotypes and we have confined our remarks in the Discussion to those where there were large and statistically significant changes.

7. Perhaps I'm missing something, but why was the vaccine coverage of individual study participants not gathered?

Response:

- The objective of the study was to examine the population impact of the vaccine not the individual effectiveness of the vaccine. From this perspective, there is no necessity for the two samples (coverage and carriage) to be undertaken in the same individuals.

8. Table 1, antibiotic use seemed to vary a lot, was this more than expected? were there circulating viruses/outbreaks at time that also may affect your analysis?

Response:

- We do not have any documented reason. These were isolated pragmatic samples that cannot be contextualised within a complex public health surveillance system as might occur in a better resourced setting.

9. Was a birth cohort analysis also done in later years that can be included to provide assurance similar results to the PCV10 survey?

Response:

- For the first two years (2016 and 2017 for Kumbotso and 2017 and 2018 for Pakoto), we did a retrospective birth cohort analysis to 'look back' to a time when we did not have any PCV10 coverage data, to estimate PCV10 coverage, We mention this as a limitation of our interpretation. For the later years, we present the observed 'cross-sectional' levels of PCV10 coverage because we collected actual data for that and thus we felt there was no need to estimate coverage using a birth cohort analysis. Additionally, the observed cross-sectional analysis provides a measure of the population PCV10 coverage (regardless of age of receipt) at the time of each carriage survey.

10. How was the 'reduction in carriage prevalence' assessed, for example if reduced from 4% to 2% is this a 50% reduction or a 2% reduction (vs change from 40% to 20%, is this 50% or 20%)? Please edit to clarify.

Response:

- We modelled prevalence ratios and ratios are relative measures of change. It should be clear that a PR of 0.5 indicates a 50% reduction, representing a relative reduction in prevalence by half, from, say, 40% to 20%.

11. There are not many studies of PCV impact in LMICs. Suggest comparing your findings against them all in the discussion to provide a complete picture, including for vaccination coverage.

Response:

- We have now made comparisons with different settings where applicable in the Discussion section.

- There are many studies from LMICs in Africa, Asia and the Americas that report PCV impact. However, because we aimed to demonstrate the impact of the vaccine programme in the overall population, comparing our findings to all PCV impact studies from LMICs may not be relevant to our interpretation. Therefore, we have limited our comparison to settings using PCV10 that have reported changes in carriage and accompanying changes in IPD. This is to support our claim for the utility of carriage as a proxy for disease in settings where surveillance may be unrealistic.

Minor issues

1. Line numbers would be helpful

Response:

- We have included line numbers

2. I don't think SRD is an informative acronym

Response:

- We have removed this acronym

3. I didn't find Figure 1 particularly informative, perhaps this timeline could be in the supplement, leaving another figure for data or a participant flow chart?

Response:

- Figure 1 gives a pictorial timelines of the carriage and coverage surveys. Reviewer 1 was keen to see the temporal relationship between vaccine introduction and the carriage surveys and this is best explained in the figure. We have opted to leave it in the main text.

4. Italicise gene names (*lytA*)

Response:

- Done

5. Tables, I think additional explanatory footnotes would be helpful to aid the reader (without having to consult text)

Response:

- We have now included footnotes in all the Tables in the manuscript and the supplement.

6. Related to above, for Table 1 specify whether current (or within timeframe X) clinical symptoms, whether household composition includes the participant

Response:

- Done

7. Table 1, was only data on solid fuel collected?

Response:

- We have included the other sources of household cooking fuel

8. Colours in Fig 4 are not informative, suggest colour VT and NVTs. If use colour, then each serotype should be a different colour so the reader can easily see changes over time.

Response:

- We have reviewed the figures and used different colours for VTs and NVTs (now Figure 2).

9. Did I miss the serotype data for the <5s?

Response:

- We have included figures for both age groups (Figure 2, Tables S2/S3).

10. Check wording in Discussion “However, NVT replacement was limited to the population ≥ 5 years in the urban site.” (limited in urban, limited to rural?)

Response:

- We have re-written this clearly now, page #11, lines 189-191

“The decline in VT carriage prevalence in Nigeria was accompanied by an increase in NVT carriage prevalence among children in Kumbotso (rural) and among older persons in both settings, with adjusted prevalence ratios of 1.26-1.34.”

11. Is there evidence that replacement serotypes were present at low abundance (e.g. by microarray)? Given that unmasking does not appear to be a major contributor in the literature, and that the relationship of PCV and density is very unclear that this stage, suggest change “likely possibility” to “possibility” when talking about unmasking/density.

Response:

- We did not explore the issue of multiple serotype carriage, serotyping just one colony per plate from each swab STGG.
- We have revised the text to avoid any unsubstantiated discussion of unmasking and density.

12. References – italicise species (and gene names if present)

Response:

- Done

Reviewer #3

1. **Page 5/Line 85ff, Figure 1 and Results.** The authors state that “baseline carriage surveys were conducted” and published (reference #13, which they provided) but they do not clearly state that the data from this prior publication was included in this current manuscript. This should explicitly state and the authors should describe if there is additional or modified data from that prior publication in the current manuscript.

Response:

- We have now clearly stated that the baseline carriage surveys have been previously published (last paragraph of introduction and first paragraph of the results) and clarified that these results are also used in the present analysis.

Page 5/Line 85ff and Discussion. The authors should acknowledge that not having pre-vaccine program baseline carriage data is a limitation as discussed in reference #1 (Flasche et al).

Response

- *We have included this limitation in the discussion page #12, lines 203-208, reproduced below:*

“The study began more than four months after PCV10 introduction, and at the baseline survey, an estimated 7%-15% of children aged <5 years had already been vaccinated. Had the baseline survey pre-dated PCV10 introduction, the measured impact would likely have been larger. The evaluation is a ‘before-after’ study which is susceptible to confounding by secular trends in VT carriage prevalence.”

2. **Page 5/Line 90ff.** The authors should provide a rationale for their sampling design. In addition, it seems that it is really a three-stage design since there was already a

selection of one rural community and one urban community, before the next stages of household census and selection and age-stratified selection from within households.

Response:

- We now indicate (introduction) that the present study built on earlier carriage surveys in these sites. Maintaining subsequent surveys in the same sites would allow evaluate changes in carriage prevalence. The original baseline survey was conducted in two sites to purposively describe one rural and one urban environment and we make this clear in the results (page #6, lines 79-81)
- To choose participants, we used a two-stage sampling procedure as follows.
 - Selection of households from a census list obtained from annual household census
 - Selection of one participant from each of the selected households in step 1 above in an age-stratified manner. This was based on the methodology previously adopted in studies in Kilifi, Kenya to ensure all ages are recruited.¹
- We have clarified the sampling strategy in the methods section, pages#16-18, lines 270-322 The statistical sampling methods were conducted in two stages, which are now clearly detailed; the third, higher, stage of sampling is inherent in the purposive selection of two exemplar sites (among thousands of possibilities in Nigeria) and this arbitrary choice is made explicit in the methods (first paragraph). Without contesting that this can be described as three selections, we still feel that it is more efficient to describe the process a selection of sites followed by a two-stage statistical sampling procedure.

3. Page 7/Line 98ff, Tables 1, S1A,S1B. The authors state that they selected participants from 10 age groups but they reported their age-specific data using just 5 age groups which contained variable proportions of the pre-defined age groups (<5 (3 age groups), 5-9 (1 age group), 10-19 (2 age groups), 20-39 (1 age group) ≥40 (3 age groups)). The actual proportions of participants from these combined age groups do not reflect equal proportions from each of the 10 pre-defined age groups. The authors should better explain their recruitment by age group sequence and how they managed

missing age groups from within household, which appears to be mainly an issue with lower proportions of adult participants.

Response:

- The small sample size suggests there will be considerable instability in prevalence estimates from within the original sampled age-strata, and considerable complexity in describing results in 10 strata of age, in two sites, and over four surveys. To improve stability and increase communication efficiency the majority of our analyses are divided into just two strata, <5 years (target age group for direct protection through vaccination) and ≥ 5 years (potential beneficiaries of indirect protection). Consistent with this approach we have now removed individual age strata from Table 1 and provide (crude) prevalence results by 10 age strata only in the supplementary figures (Figure S2). Nonetheless the stratification scheme as two significant advantages. Firstly, we were able to obtain a total population (or two stratum) prevalence with accuracy by age-standardizing age-specific prevalence estimates to the local population age structure. Secondly, we have reasonable amounts of data from across the whole span of life and these may be used to populate transmission dynamic models of the Nigerian environment to reflect variations in both prevalence of carriage and contacts with other age groups. This consideration is beyond the scope of the current paper and we have not therefore explored it here.
- Details of number recruited and response for each of the sampled age groups are already provided in the supplement (now Tables S1 and Figure S1).
- Variation in the number of participants in each age group with each survey will not inherently bias the results because they were dealt with through age-standardization. Nonetheless, variable participation could reflect differential biases that we are unable to anticipate and control for. We offer some reasons for the variation in participation below which are now included in the footnote for Table S1 as recommended by reviewer #2 above
 - Some age groups declined to be swabbed after agreeing to participate, with the main reason being believing that the procedure was too 'invasive'. This was particularly common among school-aged children and young

adolescents. To overcome this, we oversampled such age groups to ensure we get adequate numbers

- There were sometimes errors in how the ages were reported at recruitment. When this occurred, we reclassified participants into the correct/appropriate age groups at the time of interview and then resampled from the wrongfully assigned age group.
- We recruited the ten age groups in sequence from the youngest. If there was no participant in that age group, we selected the next age group in sequence and looked for missing age group in the next household. (Page #17, lines 303-306)
- If the household was known to be occupied, but there was no one at home revisited later. If the house was non-residential unoccupied or empty, we chose the next household on the list. (Page #17, lines 295-297)

4. Page 7/Line 104ff and Line 113ff and Discussion. The authors provide two different sample size estimates, one based on detection of reduction of vaccine serotype carriage and one based on estimated of vaccine coverage. This seems superfluous. The authors did not state what their primary outcome of interest was and so it is not clear which sample size estimate was more important. Regardless, they did not meet their sample size goal by either estimate for any of the surveys (100 participants in each age group in each survey or 639 children in each survey). This should be acknowledged in the discussion of limitations.

Response:

- We conducted two different surveys each year with different target populations
 - Carriage surveys (2017-2020) among resident of all ages
 - PCV coverage surveys (2018-2020) among children aged <5 years
- Therefore, we calculated sample sizes separately for the two surveys.
- A total of 1000 participants (residents of all ages) per carriage survey was the minimum sample to detect a 50% decline in VT carriage per survey from the baseline levels. (Pages #17-18, 30-3313. In Kumbotso we had a target of 5,000 (5 x 1000) and we sample, 4,679; in Pakoto we had a target of 4,000 (4 x 1000) and we sampled

3,643. These fell short of the target by 6.4% and 8.9% and the magnitude of these deficiencies is sufficiently small to discount as having significant impact on the study interpretation. The variation of recruitment by age was more variable (this point was dealt with above under Reviewer 3, point 4).

- A total of 639 participants (children aged <5 years) was the minimum sample required to assess a at least 50% PCV coverage per survey accounting for clustering at household level, assuming at least 2 eligible children per household and an ICC of 0.33. (Page #18, lines 316-322). In Kumbotso we had a target of 1,917 (3 x 639) and we sampled 2,165 children; in Pakoto we had a target of 1,278 (2 x 639) and sampled 1,313 so the sample sizes were met for these surveys.

5. Page 10/Line 170ff. The authors should state whether there were participants who took part in two or more surveys. They should also state whether there were two or more participants from individual households in any of the surveys – presumably not based on their description of the recruitment, but this should be stated explicitly.

Response:

- We have already indicated in the methods that
 - For the carriage surveys, we recruited 1 participant per household (Page #17, line 298-299)
 - For the vaccine coverage surveys, we recruited all eligible children per household (Page #18, lines 315-316).
- Since these surveys were cross-sectional nature, it is possible that participants have participated in more than one survey in different years. Participants were sampled randomly irrespective of previous participation and we did not attempt to link such participants if they existed. Given independent random samples, and given a relatively small sampling probability at each survey, there is no good reason to do so.

6. Page 10/Line 173. The authors refer to Table S1. Do they mean Table S1A and S1B? These distinct tables should be referred to separately. Also there is no Figure S1. Do they mean Figure S2A and Figure S2B?

Response:

- Thank you for pointing out this problem. We have now numbered the Tables in the order they are cited.

7. Page 12/Line 180. There is no Figure S3.

Response:

- We have now numbered the Figures correctly.

8. Page 12/Line 181ff, Table 2 and Page 15/Line 192ff and Table 3. There was a statistically significant reduction in overall pneumococcal carriage in children under 5 years of age in the urban site but not the rural site. The authors state this but do not reflect on it in the discussion. Declines in overall carriage after introduction of pneumococcal conjugate vaccine programs have been reported in other countries, particularly high income countries where the force of infection is lower (and often much lower) than in low and middle income countries. Do the authors think that there was be a significantly lower force of infection in their urban setting vs their rural setting?p

Response:

- This is an interesting, if tangential, result in the paper. The PR for the decline in overall carriage is 1.00 (0.95-1.05) in the rural setting and 0.72 (0.65-0.80) in the urban setting. Both settings see a decline in VT carriage prevalence, though this is greater in the urban than rural setting (PR 0.34 vs 0.52) but the evidence for serotype replacement is also weaker in the urban setting than the rural setting (1.03 vs 1.30). The sites were certainly (and deliberately) different² and we could speculate that the force of infection was lower in the urban setting than the rural but the decline in overall carriage in the urban site might also be attributable to the more rapid rise in vaccine uptake, and a lag in reaching an equilibrium appropriate to the new level of population immunity. The discussion is already in excess of 1,400 words (though shorter in the new version than the last) and we did not find priority to discuss this issue in this limited word count.

9. Page 20/Figure 3. The scale or percentages on the y-axis of each of the figures should be the same. As presented the y-axis on the left goes from 0-80% and on the right from 0-100%.

Response:

- We have indicated in the footnote where scales differ.

10. Page 20/Line 225. There is not Table S2.

Response:

- We have now numbered the Tables correctly.

11. Page 20/Lines 225ff, Figure 4 and Figure 5. Both Figure 4 and Figure 4 are labelled as referring to persons ≥ 5 years old in the rural sites, but the text refers to all ages and both the rural and urban sites. This is confusing and should be clarified or corrected.

Response:

- Thank you for pointing this out. We have now correctly labelled all figures.

Bibliography

1. Hammitt, L. L. *et al.* Population effect of 10-valent pneumococcal conjugate vaccine on nasopharyngeal carriage of *Streptococcus pneumoniae* and non-typeable *Haemophilus influenzae* in Kilifi, Kenya: findings from cross-sectional carriage studies. *Lancet Glob. Health* **2**, e397-405 (2014).
2. Adetifa, I. M. O. *et al.* Nasopharyngeal Pneumococcal Carriage in Nigeria: a two-site, population-based survey. *Sci. Rep.* **8**, 3509 (2018).

REVIEWER COMMENTS

Reviewer #1 (Remarks to the Author):

Nature Communications Manuscript: NCOMMS-22-1011

Title: The impact of introduction of the 10-valent pneumococcal conjugate vaccine on pneumococcal carriage in Nigeria

This is a manuscript by Adamu et al., reporting the findings from an assessment of population-level impact of PCV introduction in rural and urban sites in Nigeria on pneumococcal carriage and PCV10 vaccination coverage.

The rebuttal and changes to the manuscript are well presented and clear, thank you.

I have no further comments or concerns.

Reviewer #2 (Remarks to the Author):

Thanks to the authors for their responses and explanations. Overall I am satisfied with these, and just have some minor comments which can be considered for readability:

1. Abstract. I think the first few lines read more like an introduction, so perhaps can be shortened for the abstract (although appreciate these changes were made to clarify the purpose of the manuscript to the reviewers and for the general readership of Nat Comms)
2. Abstract. Need to edit the text about linear decreases, as this does not appear consistent with the new results for children <5.
3. I think the results for the relationships between coverage and VT prevalence re well handled, and don't want to run the risk of over interpreting. However, I was intrigued that there was a non-linear relationship with younger age groups was, especially given that linearity was seen for older age group (who have less direct effects). Was curious whether there any differences in the relationship depending on how vaccine status was recorded?
4. Discussion. I think it is reasonable and appropriate to make the case that the current study would likely be a minimal estimate/likely estimate of the effect on IPD, but wasn't convinced it would translate to "at least x%". Perhaps some softening of the language here may be appropriate? As an example, serotype 3 is rare in carriage, more common in disease, and poorly protected by PCVs, so in such a scenario vaccine impact may be overestimated by carriage.
5. Discussion. A similarly nit-picky comment: I think "may" is a better term than "would likely" (new line 70) when considering the possible effect of an earlier baseline survey. PCV may not have had substantial effects in that time especially if coverage was low.
6. Discussion. I think these results would be of interest beyond Nigeria (new line 121) so given broad journal readership, suggest adding 'and elsewhere' or similar.
7. Figure 2. The axes are hard to read, presumably these will be addressed in editing.
8. Figure 3. I found the legend a little confusing. The general statement mentions age stratification, but that only applies to 3B. I wasn't sure what the colours represented in either 3A or 3B; in the latter red seems to relate to the older age group, but not to the same thing in 3A? Also, I agree with the statement in the line in the text (new line 16-18) that the relationship for children <5 years is non-linear, but in this case would it be more appropriate to have a line of best fit/remove the linear line from Figure 3B? If I just looked at the figure I would get the impression the authors were trying to make the case that it was linear.

Reviewer #3 (Remarks to the Author):

My comments are based on a re-review of the originally submitted manuscript which has been very extensively revised and improved overall. The previous reviewer comments have generally been addressed.

A general comment about the results is that the authors only present the results of tests of some statistical significance for some comparisons, and all results presented are significant. They have not clearly stated that all other comparisons were not statistically significant.

Page 6/Lines 91-3 and Table 2. The authors state that overall and NVT carriage levels are "higher" in children <5 years, compared with persons ≥ 5 years. First, "higher" seems to be based on just a visual inspection of the proportions and 95% CI in the table and no statistical test is provided. This should be clarified. Second, what about VT carriage levels? Visual inspection of those proportions and 95% CI also appear to be higher in children <5 years, compare with persons ≥ 5 years.

Page 7/Lines 102-5 and Table 2. The authors refer to 8 prevalence trends: VT <5 years, VT ≥ 5 years, NVT <5 years, NVT ≥ 5 years, with all 4 of these trends in both of the study sites. Yet, they only provide a single statistical test result: "Chi-squared test for trend, $p < 0.001$." Was the $p < 0.001$ for each of the 8 tests, or did they somehow combine some or all of the data and conduct just a single test? This should be clarified.

Page 8/Lines 129-34 and Supplementary Tables S4 and S5. This information, on vaccine serotype (VT) carriage related to 4 additional vaccines, appears to be newly added to the manuscript. All of this is confusing and should either be better explained or removed. Were these additional vaccines used in any of the study sites? If so, more information on the frequency of their use, compared to PCV10, should be provided and all the results should be reconsidered related to the use of more than one vaccine. If none of these other vaccines were used, the text and both tables should be removed as they are not relevant to the current manuscript unless the authors want to comment on them in the discussion.

Supplementary Figure 1 could be removed as it adds no information not already provided in the text (Page 6/Lines 81-83) and Table S1.

Tables S2 and S3, on changes in serotype-specific carriage prevalence, are interesting but do not add much information to what is already presented in Table 3 and the text on Pages 7-8/Lines 114-128., and so could be removed.

REVIEWER COMMENTS

Reviewer #1 (Remarks to the Author):

Nature Communications Manuscript: NCOMMS-22-1011

Title: The impact of introduction of the 10-valent pneumococcal conjugate vaccine on pneumococcal carriage in Nigeria

This is a manuscript by Adamu et al., reporting the findings from an assessment of population-level impact of PCV introduction in rural and urban sites in Nigeria on pneumococcal carriage and PCV10 vaccination coverage.

The rebuttal and changes to the manuscript are well presented and clear, thank you.

I have no further comments or concerns.

Response:

We thank the reviewer for their time and effort.

Reviewer #2 (Remarks to the Author):

Thanks to the authors for their responses and explanations. Overall I am satisfied with these, and just have some minor comments which can be considered for readability:

1. **Abstract.** I think the first few lines read more like an introduction, so perhaps can be shortened for the abstract (although appreciate these changes were made to clarify the purpose of the manuscript to the reviewers and for the general readership of Nat Comms)

Response

- Following the comments we received from the earlier version, we thought we had not clearly communicated the scope and purpose of this work clearly. We have re-written it in this way to ensure that we state the purpose of the manuscript early on.
2. **Abstract.** Need to edit the text about linear decreases, as this does not appear consistent with the new results for children <5.

Response

- Thank you. We have edited this.
3. I think the results for the relationships between coverage and VT prevalence re well handled, and don't want to run the risk of over interpreting. However, I was intrigued that there was a non-linear relationship with younger age groups was, especially given that linearity was seen for older age group (who have less direct effects). Was curious whether there any differences in the relationship depending on how vaccine status was recorded?

Response:

- We defined vaccine status using either card or caregiver recall, and the average card retention across the surveys ranged between 70-80% and improved over time (see page 7, lines 71-74). We believe that assessing vaccination status using both card and recall is more robust. We acknowledge that caregiver recall can potentially introduce recall bias. However, because caregivers that retain vaccine cards are more likely to have vaccinated children, restricting to card only will likely overestimate PCV10 coverage and underestimate vaccine impact. A priori, our assessment was that the bias of restricting to cards would be much greater than the bias of including verbal reports and therefore we did not assess the sensitivity of the definition of vaccination status.
4. **Discussion.** I think it is reasonable and appropriate to make the case that the current study would likely be a minimal estimate/likely estimate of the effect on

IPD, but wasn't convinced it would translate to "at least x%". Perhaps some softening of the language here may be appropriate? As an example, serotype 3 is rare in carriage, more common in disease, and poorly protected by PCVs, so in such a scenario vaccine impact may be overestimated by carriage.

Response

- The statement "at least X%" is derived from the argument made in the introduction; "At the population level, PCVs provide indirect protection, regardless of vaccine status, by reducing everyone's exposure to new infections from VTs." The argument is expanded in the discussion (pages #10-11 lines 189-201). If the prevalence of serotype Y carriage is reduced by 50% in the population – then the number of individual contacts with a person carrying serotype Y are also reduced by 50%. Even if vaccine had no impact on the probability of acquisition following such a contact this would reduce the number of new infections by 50% and if every new infection has a finite probability of progression to invasive disease, the incidence of invasive disease would decline by 50%. In fact, in addition to reducing the number of opportunities to become infected, the vaccine also has direct effects among those who have been immunised; it reduces the chance that such a contact would be successful and it reduces the chance that such an infection, if it occurred, would progress to invasive disease. So the vaccine impact on the number of potentially infectious contacts is a very considerable underestimate of its total impact. In these circumstances 'at least 50%' is more than justified.
- This argument is predicated on the assumption that the incidence of invasive disease is proportional to the incidence of acquisition which was promulgated in the early 1990s following longitudinal carriage studies of children in Alabama. (Gray et al. 1980) These showed the risk of disease was temporally associated with the timing of acquisition of carriage. An alternative view, that has gained more weight in recent years, is that the incidence of disease is constant throughout a period of carriage. The argument in favour of this hypothesis is that it yields an improved model fit to empiric data. (Løchen et al. 2022) However, if the incidence of disease is directly related to the prevalence of carriage (as opposed to the incidence of acquisition of carriage) then we can use the change in carriage prevalence in the population as a whole to estimate the change in disease incidence directly. So a 50% reduction in prevalence of carriage means a 50% reduction in the carriage-days-at-risk for invasion and a logical 50% reduction in invasive disease incidence.
- The arguments do isolate a single element in a complex equation but the obvious confounders do not undermine the conclusion. For example, a reduction in transmission may simply 'delay' each person's first exposure to serotype Y and this could theoretically lead to a higher risk of disease. In

general, older children are less likely to become carriers, despite similar exposure experiences as younger children and the invasiveness index in adults is lower than in children, suggesting immune maturation against invasion with age, not all of which will be related to prior specific exposure.

5. **Discussion. A similarly nit-picky comment: I think “may” is a better term than “would likely” (new line 70) when considering the possible effect of an earlier baseline survey. PCV may not have had substantial effects in that time especially if coverage was low.**

Response:

- Thank you. We have edited the text to “may”, see page 12, line 228.

6. **Discussion. I think these results would be of interest beyond Nigeria (new line 121) so given broad journal readership, suggest adding ‘and elsewhere’ or similar.**

Response:

We agree that the results would be of interest beyond Nigeria. However, the statement in the comment above is in acknowledgment of the diversity of Nigeria with regards demographic and epidemiological factors that can influence variation in carriage burden and vaccine coverage.

7. **Figure 2. The axes are hard to read, presumably these will be addressed in editing.**

Response:

- Thank you. We will defer this to the editing stage.

8. **Figure 3. I found the legend a little confusing. The general statement mentions age stratification, but that only applies to 3B. I wasn’t sure what the colours represented in either 3A or 3B; in the latter red seems to relate to the older age group, but not to the same thing in 3A? Also, I agree with the statement in the line in the text (new line 16-18) that the relationship for children <5 years is non-linear, but in this case would it be more appropriate to have a line of best fit/remove the linear line from Figure 3B? If I just looked at the figure I would get the impression the authors were trying to make the case that it was linear.**

Response:

- We have now clearly separated the captions for Figures 3A (top graph) and 3B (bottom graph) indicating that Figure 3A is the Annual Coverage of two doses of PCV10 among children aged <5 years; and Figure 3B is the relationship in the two sites between PCV10 coverage among children aged <5 years and VT carriage stratified by age.
- We have included a point on the non-linear relationship for children aged <5 years in the discussion, Page #14, lines 272-283. We have now plotted an

exponential curve for children aged <5 years which showed a better fit to the data for this age group. We also included a graph in the supplement comparing a linear and non-linear (log-linear) model fit between VT carriage and PCV coverage (see Figure S3).

Reviewer #3 (Remarks to the Author):

My comments are based on a re-review of the originally submitted manuscript which has been very extensively revised and improved overall. The previous reviewer comments have generally been addressed.

A general comment about the results is that the authors only present the results of tests of some statistical significance for some comparisons, and all results presented are significant. They have not clearly stated that all other comparisons were not statistically significant.

Page 6/Lines 91-3 and Table 2. The authors state that overall and NVT carriage levels are "higher" in children <5 years, compared with persons ≥ 5 years. First, "higher" seems to be based on just a visual inspection of the proportions and 95% CI in the table and no statistical test is provided. This should be clarified. Second, what about VT carriage levels? Visual inspection of those proportions and 95% CI also appear to be higher in children <5 years, compare with persons ≥ 5 years.

Response:

- We did not conduct any statistical test to compare carriage prevalence across age groups because our hypothesis was focused on the change in carriage prevalence over time (see Methods section page #20, lines 389-390).
- Overall carriage prevalence has been found to be higher among children than among older persons in every setting in which it has ever been directly compared and so the hypothesis does not require testing in our dataset. As observed by the reviewer, it is possible to infer that a statistical test of difference would meet the criterion of $p < 0.05$ for all years in both sites for overall carriage (and for NVT carriage) because the 95% CIs do not overlap. This is also the case for VT carriage prevalence in the baseline survey but, because of the vaccine intervention, this age-difference in carriage prevalence becomes less apparent with time and so we could not summarise the VT carriage prevalence by age economically. The purpose of this short paragraph of results is to highlight that the epidemiology in Nigeria is conventional, and consistent with that in other settings. We have added a short sentence to affirm that VT carriage prevalence is also higher in young children 'in the baseline surveys' (page #6, lines 99-101).

Page 7/Lines 102-5 and Table 2. The authors refer to 8 prevalence trends: VT <5 years, VT ≥ 5 years, NVT <5 years, NVT ≥ 5 years, with all 4 of these trends in both of the study sites. Yet, they only provide a single statistical test result: "Chi-squared test for trend, $p < 0.001$." Was the $p < 0.001$ for each of

the 8 tests, or did they somehow combine some or all of the data and conduct just a single test? This should be clarified.

Response:

- We have now edited this and indicated the Chi-squared test for trend, $p < 0.001$ is for the 2 trends for VT change in the total population for both sites. (page #7, lines 109-111).
- The trends in NVT carriage are not significant for both sites and we have now stated this (Page #7, lines 114-117)
- Additionally, for changes in age-stratified VT and NVT carriage prevalence (Pages #7-8, lines 122-129), we also state the prevalence ratios (comparing and refer to Table 3 which also has the 95% CIs for these ratios. We believe that the readers can infer a statistical test of difference would meet the criterion of $p < 0.05$ for 95% CIs that do not include the null value of 1.

Page 8/Lines 129-34 and Supplementary Tables S4 and S5. This information, on vaccine serotype (VT) carriage related to 4 additional vaccines, appears to be newly added to the manuscript. All of this is confusing and should either be better explained or removed. Were these additional vaccines used in any of the study sites? If so, more information on the frequency of their use, compared to PCV10, should be provided and all the results should be reconsidered related to the use of more than one vaccine. If none of these other vaccines were used, the text and both tables should be removed as they are not relevant to the current manuscript unless the authors want to comment on them in the discussion.

Response:

- At present the vaccine in use in Nigeria is the GSK 10-valent PCV, and we focus our analysis on the serotypes in this vaccine. In the previous review, another reviewer asked us to comment on how the serotypes relate to the PCVs on or approaching the market. So, we included this as analysis in the supplement.

Supplementary Figure 1 could be removed as it adds no information not already provided in the text (Page 6/Lines 81-83) and Table S1.

Response:

- We agree that Figure S1 and Table S1 present similar information as the proportions presented in Figure S1 can be calculated directly from the

numbers in Table S1. However, Figure S1 is more visually accessible and there is no constraint of space in the supplement as there is in the main paper.

Tables S2 and S3, on changes in serotype-specific carriage prevalence, are interesting but do not add much information to what is already presented in Table 3 and the text on Pages 7-8/Lines 114-128., and so could be removed.

Response:

- We agree that Tables S2 and S3 are interesting and may be of more especial interest to colleagues focused on the ecological impact of PCV10 on the population structure of pneumococci. In Table 3 we only report overall and VT and NVT age-stratified carriage prevalence. In the text, we only mention the top serotypes. While in Tables S2 and S3, we report the age-stratified serotype-specific carriage prevalence for the individual PCV10, PCV13 additional serotypes and NVTs with >1 isolate. The data are not misleading and may be of more interest to some readers than others.

Reference

- Gray, B.M., Converse, G.M. and Dillon, H.C. 1980. Epidemiologic studies of *Streptococcus pneumoniae* in infants: acquisition, carriage, and infection during the first 24 months of life. *The Journal of Infectious Diseases* 142(6), pp. 923–933.
- Løchen, A., Truscott, J.E. and Croucher, N.J. 2022. Analysing pneumococcal invasiveness using Bayesian models of pathogen progression rates. *PLoS Computational Biology* 18(2), p. e1009389.